# Physiological and pathogenic T cell autoreactivity converge in type 1 diabetes

Anne Eugster [1,7], Anna Lorenc [2,6,7], Martin Kotrulev [3,4], Yogesh Kamra [2], Manisha Goel[1], Katja Steinberg-Bains[1], Shereen Sabbah [2], Sevina Dietz [1], Ezio Bonifacio [1,5], Mark Peakman [2] & Iria Gomez-Tourino [2,3,4] ✉

Autoimmune diseases result from autoantigen-mediated activation of adaptive immunity; intriguingly, autoantigen-specific T cells are also present in healthy donors. An assessment of dynamic changes of this autoreactive repertoire in both health and disease is thus warranted. Here we investigate the physiological versus pathogenic autoreactive processes in the context of Type 1 diabetes (T1D) and one of its landmark autoantigens, glutamic acid decarboxylase 65 (GAD65). Using single cell gene expression profiling and tandem T cell receptor (TCR) sequencing, we find that GAD65-specific true naïve cells are present in both health and disease, with GAD65-specific effector and memory responses showing similar ratios in healthy donors and patients. Deeper assessment of phenotype and TCR repertoire uncover differential features in GAD65-specific TCRs, including lower clonal sizes of healthy donor-derived clonotypes in patients. We thus propose a model whereby physiological autoimmunity against GAD65 is needed during early life, and that alterations of these physiological autoimmune processes in predisposed individuals trigger overt Type 1 diabetes.

Autoimmune diseases are defined as pathologies where cells are destroyed by components of the immune system through autoantigen recognition[1]. This traditional view, however, fails to distinguish between autoimmune disease and physiological autoimmunity. Controlled autoimmune responses constitute a natural defence of the human body against physiological destruction of self. For example, during neonatal tissue remodelling, organs and tissues such as pancreas and the central nervous system experience extensive cell death[2-5], accompanied by immune cell activation, to clear up the area and induce tissue repair, and followed by peripheral regulation of this physiological autoreactivity to terminate its effects[6-9]. Some models have suggested the existence of a spectrum of autoimmune processes, which extend from physiological

autoimmunity to inefficient autoimmunity and overt autoimmune disease[7].

The above is supported by many studies describing autoimmune features in otherwise healthy individuals[10-14]. Unfortunately, the characteristics that determine the limits between physiological autoreactivity and overt autoimmune disease are unclear, as the defining features of these two processes are poorly understood. In the context of Type 1 Diabetes (T1D), "beneficial" autoimmunity may play a role in this disease[6]. In this work, we focus on the cellular responses against glutamic acid decarboxylase 65 (GAD65), one of the key autoantigens in T1D[15].

GAD65 is mainly expressed in the brain and the pancreas[16] [https://www.proteinatlas.org/ENSG00000136750-GAD2], two organs

[1]Technische Universität Dresden, Center for Regenerative Therapies Dresden, Dresden, Germany. [2]Department of Immunobiology, Faculty of Life Sciences & Medicine, King's College London, 2nd Floor, Borough Wing, Guy's Hospital, London, UK. [3]Centre for Research in Molecular Medicine and Chronic Diseases (CiMUS), University of Santiago de Compostela, Santiago de Compostela, Spain. [4]Health Research Institute of Santiago de Compostela (IDIS), Santiago, Spain. [5]German Center for Diabetes Research (DZD), Paul Langerhans Institute Dresden of Helmholtz Centre Munich at University Clinic Carl Gustav Carus of TU Dresden, Faculty of Medicine, Dresden, Germany. [6]Present address: Wellcome Sanger Institute, Wellcome Genome Campus, Cambridge, UK. [7]These authors contributed equally: Anne Eugster, Anna Lorenc. ✉e-mail: iria.gomez.tourino@usc.es

that undergo intense tissue remodelling during the neonatal period[2–4]. GAD65 plays a role in autoimmune diseases related to these tissues: stiff man syndrome[17], a disease affecting the central nervous system, and T1D, which affects the pancreas. T1D is a chronic autoimmune disease where pancreatic insulin-producing β cells are selectively targeted and destroyed by the immune system, resulting in loss of glucose homoeostasis and patient death if untreated[15,18,19]. The basis of this destruction is postulated to be the loss of tolerance against self-antigens[15,18,19], with the clonal expansion of autoreactive T cells being one of the hallmarks of the disease[20]. During the last few decades, an extensive effort has been made to identify the autoantigens driving the disease (e.g. insulin, Insulinoma Associated-2 (IA-2) Zinc transporter 8 (ZnT8) and GAD65, among others), and to characterise the phenotype and T cell receptor features of autoantigen-specific T cells[15,21–23].

Several immunotherapy clinical trials in T1D have administered GAD65 with the objective of inducing immune tolerance[24–33]. However, the results of these trials have shown no or little clinical efficacy, even though immune responses were clearly induced after GAD administration[24–33]. Mechanistic analyses performed during and after the trials showed that treatment with GAD65 induced changes in CD4+ T cell populations, including cytokine release after GAD65 stimulation in vitro[24,28], but these changes did not translate into clinical efficacy, for yet unknown reasons. We have previously showed that GAD65-alum immunotherapy in T1D inadvertently expands bifunctional, potentially proinflammatory Th1/Th2 GAD65-specific CD4+ T cells[28], partially explaining the lack of efficacy observed in the trial. Importantly, we and others have extensively reported the presence of autoreactive responses against GAD65 and other T1D-related autoantigens in healthy donors (e.g[13,21,22,28,34–39]). Additionally, we also observed that T cell receptor β-chains from T1D patients display abnormal shortening, rearrangement features and increased repertoire sharing[40], and that numerous GAD65 specific Th1/Th2 clonotypes are present in those with T1D and in healthy donors[28]. Therefore, it is crucial to understand and distinguish physiological autoimmunity and autoimmune disease features of autoantigen-specific T cells in T1D. Here, we aimed to reveal the physiological *versus* pathogenic features of GAD65-specific responses, to help better inform clinical trials, as well as to further understand physiological immune responses to GAD65 in health.

To this end, we stimulate peripheral blood mononuclear cells (PBMCs) with whole GAD65 protein and compare GAD65-specific CD4+ T cell responses in health and disease. We observe that GAD65-related physiological autoimmunity is frequent, and we define T1D-specific phenotype fingerprints in the T-cell compartment. By combining parallel sequencing of single-cell GAD65-specific TCRs and bulk peripheral CD4+ T cell subsets we find that GAD65-specific T1D clonotypes are more often public and convergent. Importantly, HD peripheral repertoires are populated by some exclusive GAD65-specific clones of high sizes, which include potential regulatory phenotypes, that are missing in T1D patients. Our results indicate that features of physiological and pathogenic autoimmunity against GAD65 converge in T1D, highlighting the importance of performing orthogonal and multilayered analysis of GAD65 responses in clinical trials, as well as including healthy donors as a reference to be able to discriminate physiological responses from disease- and/or treatment- specific ones.

## Results

### GAD-specific CD4+ T cells exist in similar proportions in Type 1 Diabetes patients and healthy donors

To assess the whole breadth of GAD responses in an epitope-agnostic way, we stimulated fresh PBMCs from 40 newly diagnosed T1D patients (3.2 ± 2.4 months since diagnosis) and 36 healthy donors (HD) with whole GAD protein and performed IFN-γ and IL-10 ELISPOT (Supplementary Fig. 1 and Supplementary Data file 1). There were no

differences in age, sex, or frequency of diabetogenic *DRB1*0301* (DR3) or *DRB1*0401* (DR4) haplotypes between patients and HD (Supplementary Table 1. Fisher, chi-square or Mann Whitney U tests, all ns). Unexpectedly, and contrary to what has been described for other T1D autoantigens[22], T1D patient responses were not polarised towards increased IFN-γ secretion, and both HD and T1D patients showed responses of comparable magnitude and type (Fig. 1 a–c and Supplementary Fig. 2a, b), with 42% HD and 40% T1D patients not secreting IFN-γ or IL-10 after GAD stimulation (Fig. 1c). This indicates that GAD responses exist both in health and disease, and that quantification of IFN-γ and IL-10 responses alone are not sufficient to recapitulate the whole spectrum of immune responses against GAD.

Previous reports by us and others showed that the CD154/CD69 activation marker induced (AIM) assay enables efficient identification of antigen-specific CD4+ T cells, avoiding biases inherent to other approaches such as proliferation, ELISPOT or tetramer stain[41–44]. Therefore, we stimulated fresh PBMCs from 11 newly diagnosed T1D patients and 10 HD (Supplementary Fig. 1 and Supplementary Data File 1) with whole GAD protein overnight and analysed the frequency and surface phenotype of CD154+ CD69+ GAD-specific CD4+ T cells (Supplementary Fig. 3). There were no differences in age, sex, or frequency of DR3 or DR4 haplotypes between patients and HD in this subset of donors (Supplementary Table 1. Fisher, chi-square or Mann Whitney U tests, all ns). As expected, the percentage of CD154+ CD69+ cells was significantly higher in GAD-stimulated PBMCs than in those treated with diluent alone (Supplementary Fig. 2c, d), and cell lines generated from sorted CD154+ CD69+ cells specifically responded to the antigen against which they were generated (Supplementary Fig. 4).

Surprisingly, every individual tested showed, to a bigger or lesser extent, responses against GAD, measured as percentage of GAD-specific CD154+ CD69+ cells (GSCs) (Fig. 1D and Supplementary Fig. 2c, d), with no differences between HD and T1D patients (Fig. 1d, and Supplementary Table 2. Mann Whitney U or unpaired t tests). There were no significant correlations between GAD autoantibodies, or DR3/DR4 positivity, and percentage of GSCs.

We further characterised the GSCs classifying them into true naïve (TN. CD45RO^neg CD27+ CD95^neg), central memory (CM. CD45RO+ CD27+), effector memory (EM. CD45RO+ CD27^neg), stem cell-like memory T cells (Tscm. CD45RO^neg CD27+ CD95+) or non-terminated memory T cells (NTEM. CD45RO^neg CD27^neg) (Supplementary Fig. 3). Around half of the GSCs were of CM phenotype, followed by TN and EM phenotypes (Fig. 1e), showing similar distributions to those of Staphylococcal Enterotoxin B (SEB) -induced CD154+ CD69+ cells (Supplementary Fig. 5a). As expected, we observed an enrichment in CM and EM phenotypes when compared with unstimulated cells from the same donors and blood sample (compare Fig. 1e and Supplementary Fig. 5b), highlighting that this ex vivo assay can effectively enrich for GAD-specific memory cells, while also capturing TN cells recently activated by antigen.

Overall, the surface phenotype of GSCs was not significantly different between T1D patients and HD (Fig. 1e); however, the preponderance of each phenotype varied in individuals with high *versus* low GSC frequencies (Supplementary Fig. 5C). The frequency of GSCs negatively correlated with the percentage of true naïve GSCs (Supplementary Fig. 5D), indicating that in individuals with low frequencies of GSCs, cells were more often of TN phenotype. As a comparison, we also analysed cytomegalovirus (CMV)-specific cells- these were rarely of TN phenotypes (Supplementary Fig. 5E).

In summary, our ex vivo assays showed that GAD responses are ubiquitous and present in both HD and T1D patients. Also, the magnitude of the response is related with the naïve/memory surface phenotype of GSCs, and many GAD-responsive individuals would appear as unresponsive in conventional IFN-γ/IL-10 ELISPOT assays.

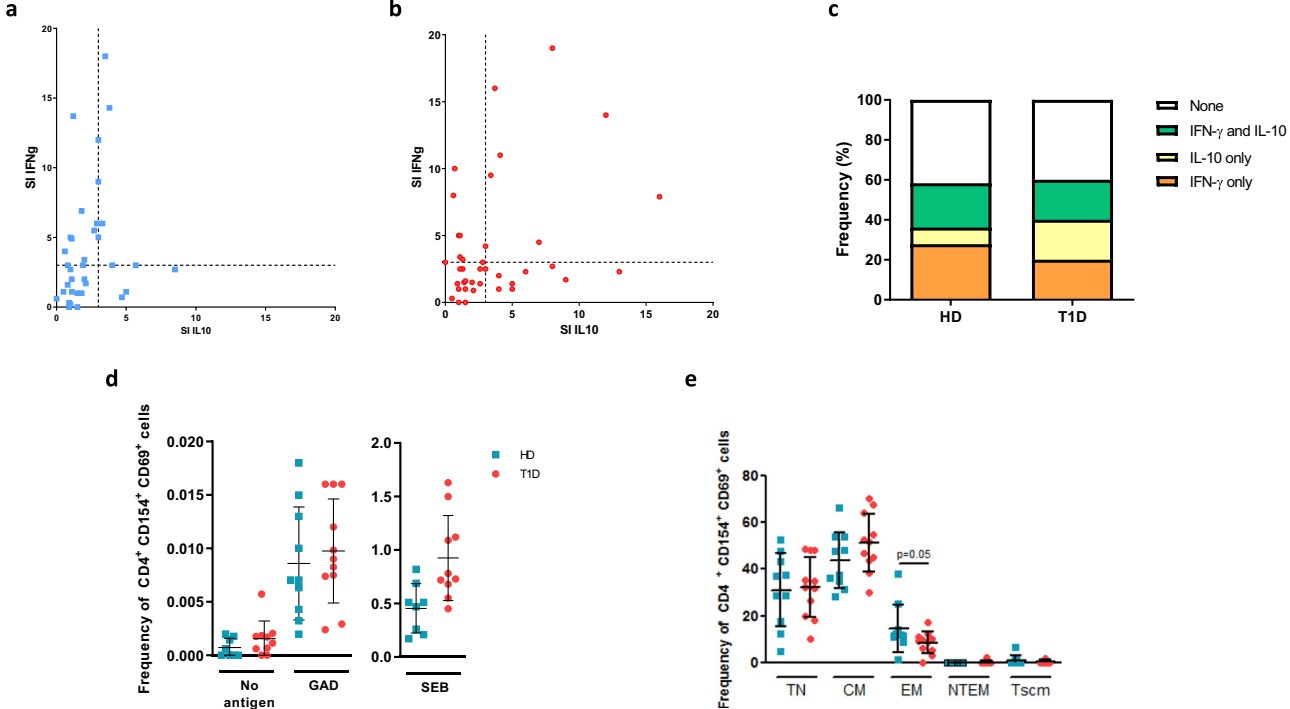

**Fig. 1 | GAD-specific responses are detected in T1D patients and healthy donors through different orthogonal methodologies. a, b** Scatter plot for IFN-γ and IL-10 ELISPOT stimulation indexes (SI) against GAD in HD (**a**) and T1D patients (**b**). Dashed lines represent the thresholds for positivity (SI = 3). **c** Classification of ELI-SPOT responses. Shown are percentages of total. Two-sided chi square test ($p > 0.05$). **d** Frequency of CD154+ CD69+ cells from 10 HD and 11 T1D patients after no stimulation ("no antigen"), stimulation with GAD or with SEB. Two-sided Mann Whitney U or unpaired t test. **e** Phenotype of GAD-specific CD154+ CD69+ cells from 10 HD and 11 T1D patients (Two-sided Mann Whitney U test. $p > 0.05$). TN: true naïve. CM: central memory. EM: effector memory. Tscm: stem cell-like memory. Blue: HD. Red: T1D patient. Error bars represent standard deviations.

## GAD responses in T1D are characterised by a combination of altered physiological autoimmune features and disease-specific phenotypes

Given that GAD-specific responses are ubiquitous, we set out to investigate GSCs at the single cell level. We performed single-cell qPCR on GSCs from 7 T1D patients and 5 HD (Supplementary Fig. 1 and Supplementary Data File 1). There were no differences in age, sex, or frequency of DR3 or DR4 haplotypes between patients and HD in this subset of donors (Supplementary Table 1. Fisher, chi-square or Mann Whitney U tests, all ns). We single cell index sorted CD4+ CD154+ CD69+ cells and classified, *a posteriori*, each single sorted cell as TN, CM, EM or Tscm (Supplementary Fig. 3). Single cells were subjected to single-cell PCR (for TCR sequencing) and microfluidics single cell quantitative PCR (for expression of immune-related genes) (Supplementary Fig. 1).

We performed dimensionality reduction of normalised and batch-corrected gene expression data for 1460 GSCs, and clustered cells in 10 groups based on gene expression similarity (Fig. 2a, b and Supplementary Fig. 6). Clustering was neither biased by qPCR run nor by individual (Supplementary Fig. 7a–d). Genes expressed differentially among clusters were determined with the hurdle model (see "Methods", Supplementary Data File 2 and Supplementary Fig. 8).

GAD responses were surprisingly diverse, including easy-to-categorise cell subsets, such as Th1 or Th2, as well as other, more complex, phenotypes (Fig. 2b). We did not observe a statistically significant differential distribution of HD and T1D cells throughout the clusters (Fig. 2c and Supplementary Fig. 7b. $p > 0.05$ in all cases, Wilcoxon test). However, we find T1D-specific differences in the expression levels of specific genes within specific clusters.

Cluster #6 and #9 constitute highly activated, cytokine secreting, proliferating memory Th1-like and Th2-like cells, respectively (Fig. 2b and Supplementary Fig. 6); these clusters, together with cluster #3 (TGF-β+ IL-2+/- TNF+/- highly activated cells) and cluster #1 (hyporesponsive cells) constitute the only four clusters with no differences in gene expression levels between T1D and HD cells. Therefore, these constitute physiological responses against GAD that are not altered in the disease. In all other clusters, T1D cells present specific features when compared to HD ones.

Cells from cluster #4, whereas similar to cluster #9 (Fig. 2b and Supplementary Fig. 6), lack expression of *IL-4* or *IL-13*, and *GM-CSF* and *IL-21* are rarely co-expressed (Supplementary Fig. 8a). GM-CSF single positive cells express significantly higher levels of *Egr2* (Supplementary Fig. 9a), suggesting a better predisposition for clonal expansion[45]. T1D cells in this cluster tend to be more often GM-CSF single positives (Supplementary Fig. 9b), and we have previously shown that GM-CSF-producing CD4+ T cells are frequent in children with T1D[46,47]. GM-CSF+ IL-21+ T1D cells express significantly lower levels of *CTLA-4* than their HD counterparts (Fig. 2d), suggestive of a lower capacity of inhibition of these responses.

Helios (coded by the gene *IKZF2*) is a marker of thymus-derived Treg cells[48,49], and most CD25hi CD127lo FOXP3+ cells express *IKZF2*[50]. We observed a group of Helios+ cells (cluster #10) (Fig. 2b and Supplementary Fig. 6). However, the majority of cluster #10 cells are FOXP3neg. Interestingly though, they bear other characteristics of Treg cells, such as *CTLA-4*, *ICOS* and *TGF-β* expression (Fig. 2b)[48,51,52]. T1D cells in this cluster express significantly higher levels of *IKZF2* than their HD counterparts (Fig. 2e) and, although low cell numbers constrain statistical analysis, there are more FOXP3+ cells within HD than T1D patients (4/6 cells *versus* 1/22, respectively).

Cluster #8 comprises memory and true naïve cells with a significantly higher than average expression of *CXCR5* (Fig. 2b and Supplementary Fig. 6). This, together with the fact that (i) IL-2 levels are higher in CXCR5 single-positive (SP) cells (Supplementary Fig. 9c), (ii)

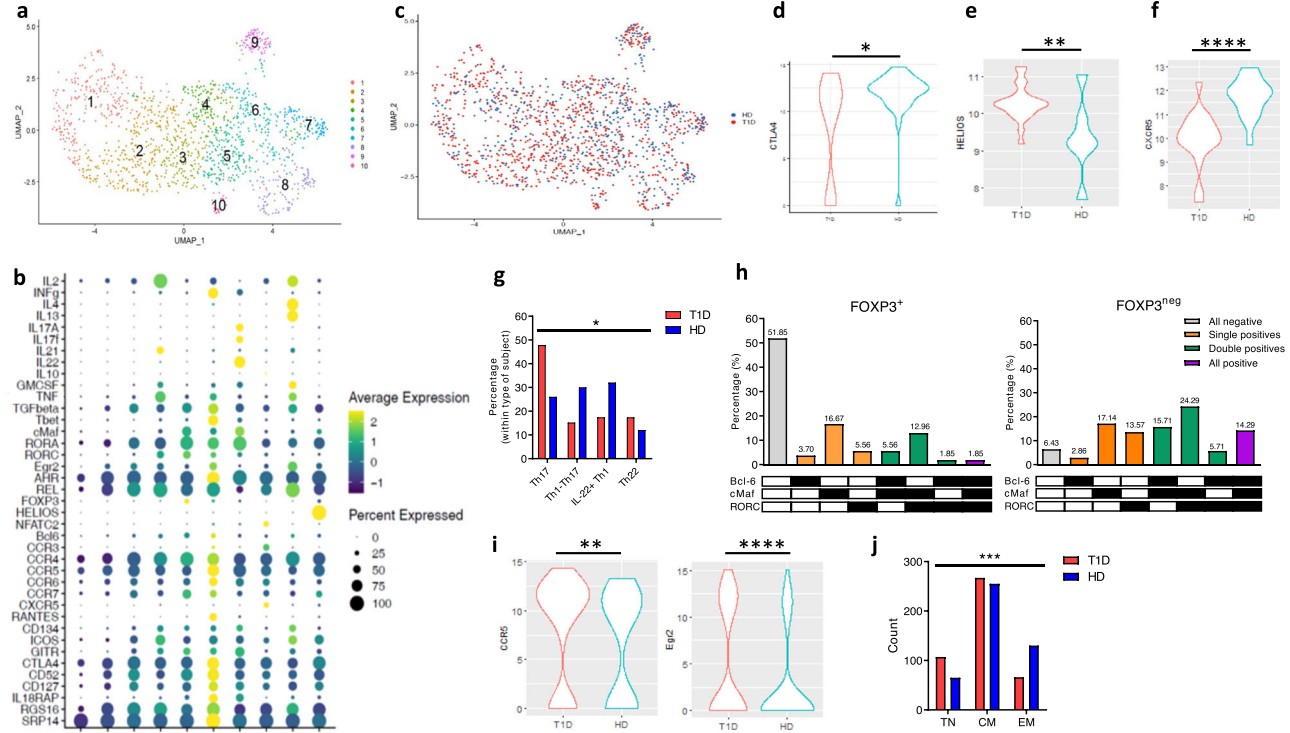

**Fig. 2 | GAD-specific cells show qualitative alterations in T1D patients. a** UMAP plot for all GAD-specific cells coloured per cluster. **b** Details of gene expression per cluster, showing the fraction of cells expressing a gene (dot size) and their expression level (colour). **c** UMAP plot coloured per type of donor. **d** *CTLA-4* expression levels in cluster #4 (two-sided Mann Whitney U test with Bonferroni multiple comparisons correction. $p = 0.043$). **e** *IKZF2* expression levels in cluster #10 (two-sided unpaired t test with Welch's correction. $p = 0.0034$). **f** *CXCR5* expression levels in cluster #8 (two-sided Mann Whitney U test with Bonferroni

multiple comparisons correction. $p = 1.97 \times 10^{-5}$). **g** Distribution of HD and T1D cells within cluster #7 subclusters (two-sided Chi-Square test. $p = 0.047$). **h** Frequencies of transcription factor-based cell subsets in FOXP3+ and FOXP3neg cells of cluster #5. **i** *CCR5* and *Egr2* expression levels in cluster #2 (two-sided Whitney U test with Bonferroni correction. $p = 2.7 \times 10^{-3}$ and $p = 1.4 \times 10^{-6}$ respectively). **j** Cell surface phenotype distributions (two-sided Chi-square test. $p < 0.0001$). **c, d, e, f, i** blue, HD; red, T1D. **g, j** white, HD; black, T1D. *: $p < 0.05$. **: $p < 0.01$. ***: $p < 0.001$. ****: $p < 0.0001$.

CXCR5 SP cells are quite often Bcl-6neg and CCR7lo (Supplementary Fig. 9d) (germinal centre Tfh cells downregulate *Bcl-6* when migrating to blood[53–55]), and (iii) CXCR5 SP cells are preferential of memory phenotype (Supplementary Fig. 9e) suggests that these CXCR5+ cells might constitute (blood) circulating activated memory Tfh cells. *CXCR5* expression levels are significantly lower in CXCR5+ T1D cells ($p = 1.97 \times 10^{-5}$. Figure 2f), even though the proportions of CXCR5+ cells between HD and T1D patients are comparable (Supplementary Fig. 9F).

Cluster #7 are activated, memory, non-proliferating *IL-22* expressing cells (Fig. 2b, Supplementary Fig. 6, and Supplementary Fig. 7e, f), that could be clearly classified into Th17-like (36.5%), Th1-Th17 (22.9%), IL-22+ Th1 (25.0%) and Th22 cells (14.6%) (Supplementary Fig. 9g). Therefore, these cells represent several facets of the Th1-Th17-Th22 spectrum. T1D cells in this cluster are more often of Th17-like phenotype, contrary to HD cells (Fig. 2g).

Cluster #5 cells present higher-than-average levels of *FOXP3* expression and are characterised by transcription factor (TF) heterogeneity, co-expressing *Bcl-6, cMaf, RORA, RORC* and/or *FOXP3* (Fig. 2b and Supplementary Fig. 6). 27.8% of cluster #5 cells are FOXP3+ and express significantly lower levels of *CD127* than their FOXP3neg counterparts (Supplementary Fig. 9h). Within FOXP3+ cells, 52% of them are triple negative for *Bcl-6/MAF/RORC* (Fig. 2h). Of those FOXP3+ cells coexpressing other TF, they are more often positive for *MAF* in several combinations (Fig. 2h), and c-Maf is required for Treg cells to secrete IL-10 [56–60]; therefore, this FOXP3+ population could constitute Treg cells, a portion of which is transitioning through specialisation programme(s). Cluster #5 FOXP3neg cells also show TF heterogeneity, but to a much higher extent (Fig. 2h), expressing higher levels of *Bcl-6, MAF* and *RORC* than their FOXP3+ counterparts (Supplementary Fig. 9h).

T1D cells in this regulatory cluster express significantly lower levels of *RORC* (Supplementary Fig. 9i).

Cluster #2 are activated cells with low cytokine and TF expression levels (Fig. 2b and Supplementary Fig. 6), presenting the highest proportions of TN cells among all clusters (Supplementary Fig. 7e,f). The expression levels of *CCR5* and *Egr2* (implicated in anergy establishment[45]) are significantly higher in T1D cluster #2 cells (Fig. 2i).

We were intrigued by TN cells being identified during the AIM assay- reassuringly, these TN cells express, as expected, less cytokines than CM and EM cells (Supplementary Fig. 10A-C). Interestingly, many of these TN GSCs are in a proliferating state, as measured by IL-2 expression, and show different characteristics depending on the cluster; for example, TN cells in the Treg/TF heterogeneous and Th1-like clusters (#5 and #6) are FOXP3+ (Supplementary Fig. 10a). Interestingly, we observed a significant increase in the proportion of TN cells in T1D patients (Fig. 2j).

Therefore, GAD-specific responses are unexpectedly diverse, detecting cytokine-secreting cells in both HD and T1D patients. These responses are equally frequent in both types of individuals, although T1D patients present disease-specific features in gene expression levels. Besides, T1D GSCs are more often of TN phenotype.

## GAD-specific TCR clonotypes are clonally expanded among different genex clusters

To address the identity and relationships among these GAD-specific cells we sequenced their T cell receptors (TCR): we obtained a total of 345 productive unique TCRB CDR3 sequences, 107 productive unique TCRA CDR3 sequences, and paired TCRA/TCRB sequences for 114 cells. This sequencing success rate is within the range of our previous studies[28] and

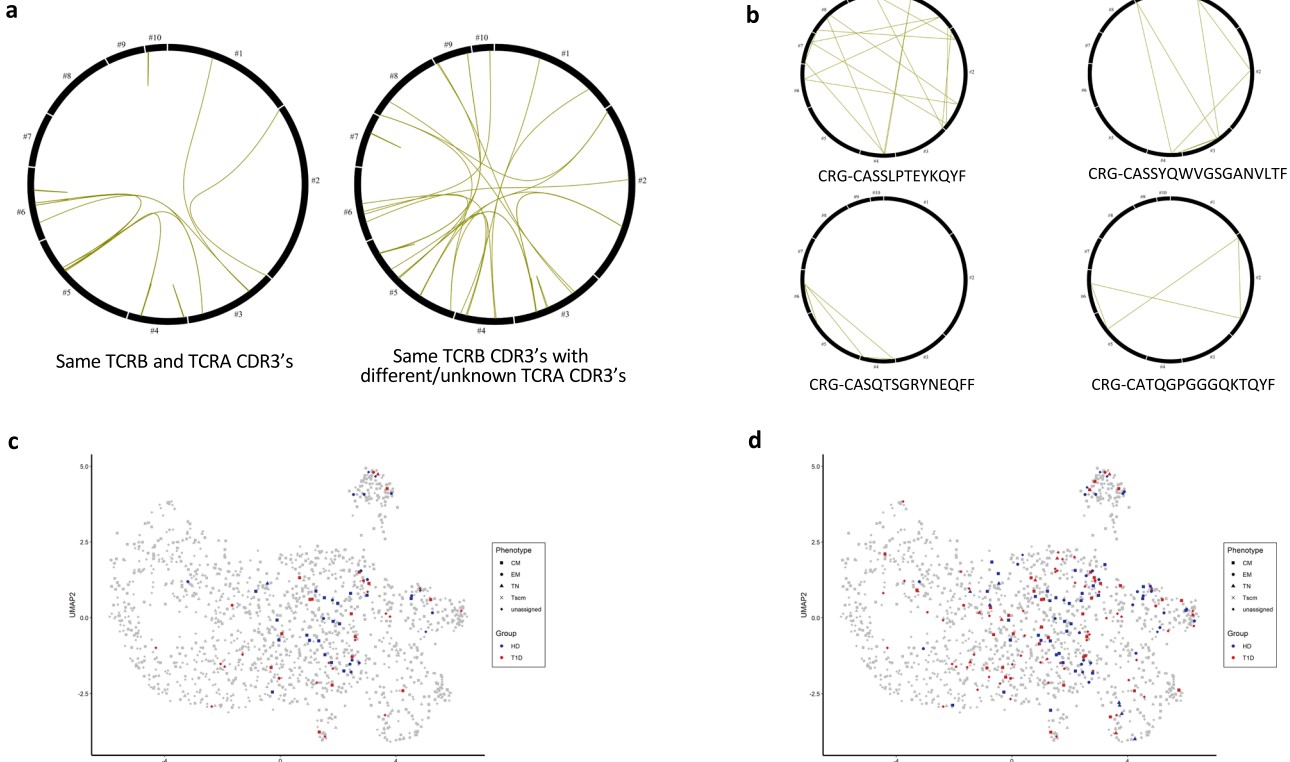

**Fig. 3 | GAD-specific clonotypes are located in multiple gene expression clusters and found in peripheral immune repertoires. a** Circus plot with lines joining cells with identical TCRB and TCRA nucleotide sequences (left) or identical TCRB CDR3 (and unknown or different TCRA) (right). **b** Circus plots with lines joining cells with similar TCRB CDR3 amino acid sequences (belonging to the same convergence group as per GLIPH algorithm, see Supplementary Table 3), and found in different donors. **c** Cells whose TCRB CDR3 nucleotide sequence is tracked back in peripheral immune repertoire subsets. **d** Cells whose TCRB CDR3 amino acid sequence is tracked back in peripheral immune repertoire subsets. **c**, **d** blue: HD cells. Red: T1D cells.

those of others[37]. Productive sequences were evenly distributed throughout our gene expression based (genex) clusters (Supplementary Fig. 11a), except for lower representation of cluster #1, identified above as comprising many naïve and lowly active cells (Supplementary Fig. 11a). Cells with both TCRA and TCRB sequences are equally distributed throughout the UMAP space for HD and T1D patients, with some enrichment of the latter in cluster #3 (Supplementary Fig. 11b).

60 cells had an identical nucleotide TCRB CDR3 sequence to some other cell(s) in the population, with 18 cells having both the same TCRB and TCRA CDR3. All individuals showed some degree of clonal expansion, with no differences between T1D patients (52%) and HD (48%). Overall, there were 23 expanded clonotypes (Fig. 3a). Of those, only 22% were constituted by cells located in the same genex cluster. The remainder expanded clonotypes were located in several clusters, indicating that the cells originated from a common precursor and then followed different differentiation paths (Fig. 3a). Some of these clonotypes appear in the regulatory cluster (cluster #5).

We then applied GLIPH[61] to identify groups of TCRB CDR3's of different amino acid sequences but potentially common specificities. The algorithm identified thirteen TCRB groups. Four GLIPH groups contained cells from more than two individuals (Fig. 3b and Supplementary Table 3), and therefore represent TCRs likely recognising the same epitope across several donors. The remainder 9 groups were constituted of two cells each from the same donor, potentially linking to individual-specific expansions.

Overall, we observed TCR expansions in GSC's, being particularly evident in clusters #6 (Th1), #9 (Th2), #4 (Th-GM/IL-21), #5 (Treg/TFhet) and TGF-β expressing cluster #3. Some TCRs present different amino acid sequences but share features that make them highly likely to recognise similar epitopes.

## GAD-specific nucleotide TCRB CDR3 sequences are public in true naïve cell subsets

We next set out to investigate the presence and relevance of these GAD-specific TCRB clonotypes in peripheral blood. For that, we sequenced TCRB CDR3 regions from bulk-sorted conventional TN, CM and Tscm CD4⁺ T cells and Tregs from the original blood draw from each donor (sequencing results available in ref. 40). Additionally, we also included TCR repertoires from other HD and T1D individuals, not analysed for GSCs here, extending the dataset to a total of 94 repertoires and 31.5 million TCRB CDR3 sequences[40] (Supplementary Fig. 1 and Supplementary Data File 1). There were no differences in age, sex, or frequency of DR3 or DR4 haplotypes between patients and HD in this subset of donors (Supplementary Table 1. Fisher, chi-square or Mann Whitney U tests, all ns).

To better understand the origin of these GSCs, we first analysed the potential publicity of GAD-specific TCRB CDR3 nucleotide sequences. Recombination events in the thymus (including insertions and deletions) are considered to be random and, as such, the probability of two individuals bearing exactly the same TCRB CDR3 nucleotide sequence just by chance is extremely low.

TCRB CDR3 nucleotide sequences from some clusters were tracked back more often than others (Fig. 3c). The average frequency of tracking back of nucleotide GAD-specific TCRB CDR3 was of 20% ± 13% (Supplementary Table 4). The frequencies of tracking back were comparable between HD and T1D patients (Supplementary Fig. 11c). As a comparison, we also single-cell sequenced CMV-specific cells from three donors. We obtained 90 productive unique CMV-specific TCRB CDR3 nucleotide sequences and the average frequency of tracking back was higher, of 36% ± 7% (Supplementary Table 4).

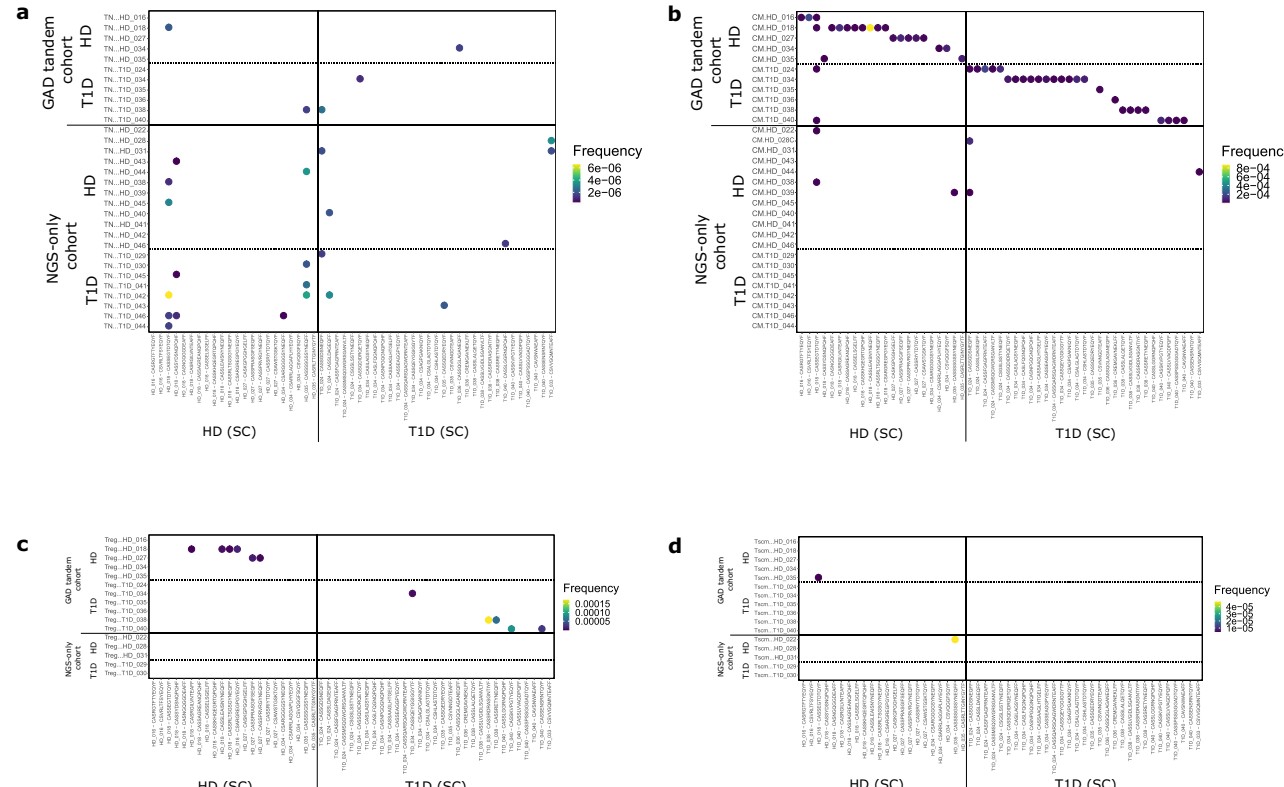

**Fig. 4 | GAD-specific TCRB CDR3 nucleotide sequences can be tracked back into specific peripheral immune cell subsets.** We browsed the GAD-specific TCRB CDR3 nucleotide sequences into 94 peripheral immune cell repertoires: 31 TN (**a**), 31 CM (**b**), 16 Treg (**c**) and 16 Tscm (**d**). We classified these repertoires into GAD tandem cohort (if both single-cell and deep sequencing took place) and NGS-only cohort (if we only performed deep sequencing). Colour represents the frequency of the given clonotype in peripheral immune cell repertoires.

We observed that, as expected, the majority of GAD and CMV TCRB CDR3 nucleotide sequences found in the CM, Treg and Tscm pools were ultraprivate, being tracked back only into the same donor in whom they were originally identified (Fig. 4b–d and Supplementary Fig. 12). CMV clonotypes found in TN pools were also ultraprivate (Supplementary Fig. 12a). Strikingly, GAD-specific clonotypes found in TN subsets are public, with 6 out of 11 clonotypes being found in more than one individual (Fig. 4a). Considering that we were looking at identity at the nucleotide level, this is suggestive of a similar recombination/selection thymic process favouring the generation of these GAD-specific clonotypes, which is not evident for CMV-specific ones (Supplementary Fig. 12a). Interestingly, for some clonotypes publicity seems to propagate into other immune cell subsets, as some clonotypes public in TN cells are also public in CM cells (Supplementary Fig. 13), and this publicity is more apparent for GAD than for CMV-specific clonotypes (compare Fig. 5a and b). The specificity of two of these public nucleotide clonotypes has been confirmed by TCR transductions (Supplementary Fig. 14).

We also observed that a higher fraction of GAD-specific clonotypes were found in Treg subsets, contrary to what we found for CMV clonotypes (compare Fig. 4c and Supplementary Fig. 12c), suggestive of an ongoing and peripherally derived regulatory response against GAD, but not CMV. The opposite is true for Tscm, where we found more CMV- than GAD-specific clonotypes (compare Fig. 4d and Supplementary Fig. 12d), aligning with CMV being a chronic infection and inducing CD4+ Tscm cells as a part of memory responses.

To address the relevance of antigen-specific TCRB CDR3 nucleotide sequences in blood, we analysed their frequency in the peripheral immune cell subsets of the same donor. Antigen-specific clonotypes were of low frequency in TN and Tscm subsets of the same donor (Fig. 5c). Interestingly, the frequencies of GAD- and CMV-specific

clonotypes in CM subsets were comparable, indicating that GAD-specific TCRB CDR3 are almost as frequent as CMV ones in CM repertoires (Fig. 5c).

Unexpectedly, we also observed that the frequency of T1D GAD-specific TCRB CDR3 nucleotide sequences in the Treg subsets of the same patients were significantly higher than those from HD or CMV (Fig. 5c. $p < 0.05$. Kruskall-Wallis + Dunn). In most instances, when a clonotype was found in the Treg subset it was also found in the CM population of the same donor (Fig. 4b, c), suggesting the presence of both Treg and conventional CM cells with the same antigen specificity and TCRB CDR3 nucleotide sequence, strongly suggesting that they originated from the same progenitor cell. Interestingly, in these cases the ratio of frequencies Treg/CM was significantly higher in T1D patients (Fig. 5d. $p < 0.05$, Mann-Whitney U test).

In summary, we found that GAD-specific nucleotide TCRB CDR3 sequences are surprisingly public in TN cells from different individuals, suggesting a similar thymic process biased towards the generation of these clonotypes. GAD clonotypes are frequent in the periphery when compared with CMV ones. Additionally, T1D patients present expansions of some GAD-specific TCRB CDR3 nucleotide sequences in Treg subsets, surpassing the frequency of their CM counterparts.

## T1D-derived GAD-specific clonotypes are frequent in HD, while HD-derived ones are rare in patients

To address the functional relevance of our antigen-specific clonotypes, we searched their amino acid sequences in the peripheral immune repertoires. We found 147 GAD-specific clonotypes in these repertoires. These tracked clonotypes originated from all genex clusters (Fig. 3d), and the average frequency of tracking back was of 44 % ± 13% (Supplementary Table 4). The frequencies of tracking back were

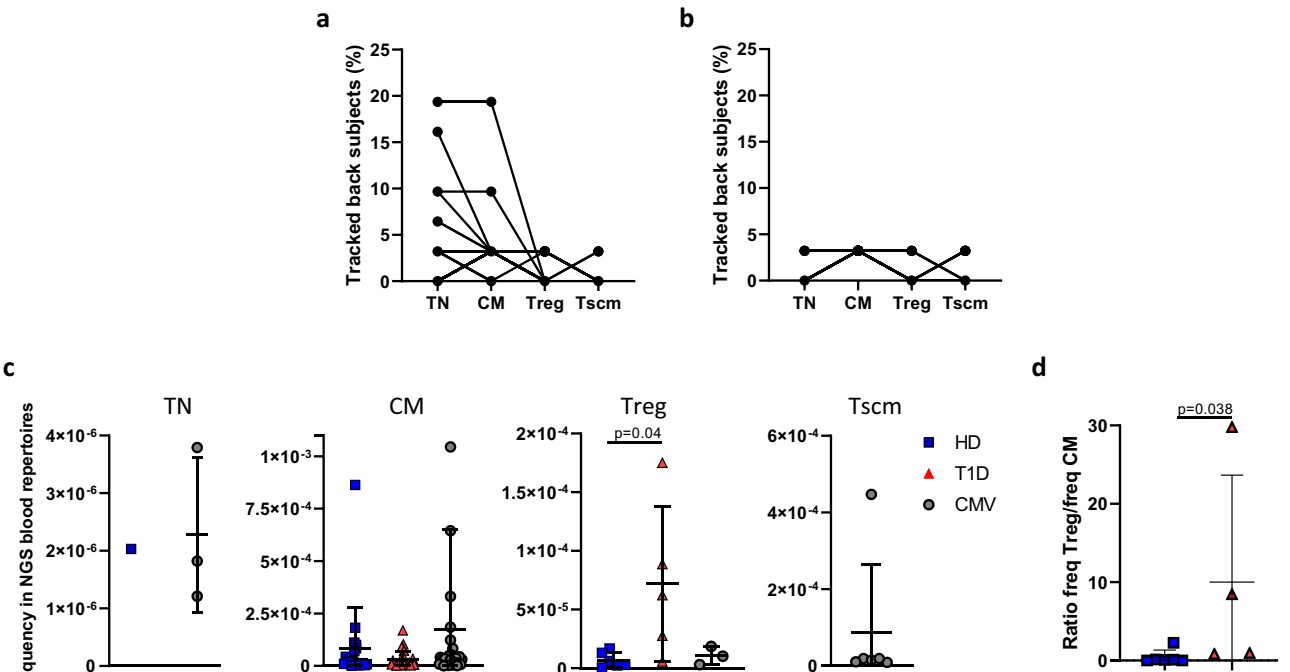

**Fig. 5 | GAD-specific TCRB CDR3 nucleotide sequences are public and frequent in specific peripheral immune cell subsets. a, b** Percentage of individuals where GAD- (**a**) and CMV- (**b**) specific TCRB CDR3 nucleotide sequences are tracked back. Although each line represents a clonotype, note that many clonotypes show the same behaviour and, as such, they are overlapped in this representation. **c** Intraindividual frequencies of GAD- and CMV- specific TCRB CDR3 nucleotide sequences (GAD from HD: 1 in TN, 19 in CM, 6 in Treg. GAD from T1D patients: none in TN, 26 in CM, 5 in Treg. CMV: 3 in TN, 32 in CM, 3 in Treg, 6 in Tscm) found in the peripheral immune repertoires (Kruskal Wallis+Dunn). One CMV clonotype

exceptionally frequent in the CM subset is not depicted in the figure to ease the visualisation of all other clonotypes (frequency=0.002529). **d** Ratios of the frequencies of GAD-specific TCRB CDR3 nucleotide sequences in Treg and CM subsets from the same donor (two-sided Mann-Whitney U test). **c, d** Blue squares: GAD-specific TCRB CDR3 nucleotide sequences found in HD. Red triangles: GAD-specific TCRB CDR3 nucleotide sequences found in T1D patients. White circles: CMV-specific TCRB CDR3 nucleotide sequences. Error bars represent standard deviations. *: $p < 0.05$.

comparable between HD and T1D patients (Supplementary Fig. 11d), and the average tracking back for CMV was of 73% ± 17% (Supplementary Table 4).

While tracking at the nucleotide level was mostly intra-individual (except for TN subsets), we found both intra and inter-individual tracking at the amino acid level in all four cell subtypes (compare Fig. 4 and Figs. 6,7). CMV-specific clonotypes were also found in all four cell subtypes, although it became apparent that they were found less often in Treg pools than GAD-specific clonotypes (compare Fig. 7a and Supplementary Fig. 15c)- as we also observed this phenomenon at the nucleotide level, it is tempting to speculate that an ongoing and peripherally derived regulatory response is more required for GAD than for CMV.

Publicity was relatively high and varied depending on the clonotype and the immune cell subset (Figs. 6,7). To further analyse publicity, we classified the antigen-specific clonotypes as extremely public, public, private or ultra-private, based on the number of donors where each clonotype was found (see "Methods"). We found 13 extremely public GAD clonotypes in the TN subsets, while only 5 clonotypes presented this extreme degree of convergence in CM, and none in Treg and Tscm (Figs. 6,7 and Fig. 8a, b). This TN-specific extreme publicity is propagated into other immune cell subsets for 5 GAD-specific clonotypes (Fig. 8a, b), suggesting that the causes of this extreme publicity are similar thymic and selection events, which are followed in some cases by expansions in memory pools. Most CMV clonotypes were of private or ultraprivate nature (Supplementary Fig. 16A).

GAD-specific clonotypes from T1D patients were found in more TN cell repertoires than HD GAD-specific clonotypes (Fig. 8c. $p < 0.05$, Mann Whitney U test). T1D patients presented higher proportions of public clonotypes in TN and CM pools than HD (Fig. 8a, b and

Supplementary Fig. 16b, c), but more private ones in the case of Treg subsets (Fig. 8d. $p = 0.02$. Chi square test). For the Tscm repertoires fewer clonotypes were tracked back and, if so, they were mostly of private or ultraprivate nature (Supplementary Fig. 16d).

Given that a clonotype can be public (i.e. present in several individuals) but rare (i.e. of low frequency when compared with other clonotypes), and to address the relevance of GAD-specific clonotypes in blood, we calculated the average frequency of all convergent nucleotide sequences and analysed the frequency distributions of T1D-derived and HD-derived GAD-specific clonotypes in peripheral immune cell subsets.

We first wondered whether highly convergent clonotypes are also highly frequent. For that we calculated the correlation between the number of convergent nucleotide sequences and the peripheral frequency of each given amino acid clonotype. We observed a high positive correlation in TN subsets (Fig. 8e, f. GAD: R = 0.96. $p < 0.0001$. CMV: R = 0.99, $p < 0.0001$. Spearman correlation) suggesting that, as expected, the ease of being generated and selected is determinant for a clonotype to be frequent in TN pools. Correlation values are comparable between HD-derived and T1D-derived GAD clonotypes (0.98 and 0.95, respectively. Figure 8e). We also observed that a high number of T1D-derived GAD clonotypes show medium frequencies and convergences, contrary to what happens for HD-derived and CMV ones (Fig. 8e, f).

In CM subsets, this correlation between convergent nucleotide sequences and frequency is less strong (GAD: R = 0.78. $p < 0.001$. CMV: R = 0.68. $p < 0.001$. Spearman. Figure 8g, h), being relatively higher in T1D patients (R = 0.83) than HD (R = 0.72). In the case of HD-derived GAD clonotypes and CMV ones, the majority of the clonotypes have low convergence associated with variable frequencies, explaining this

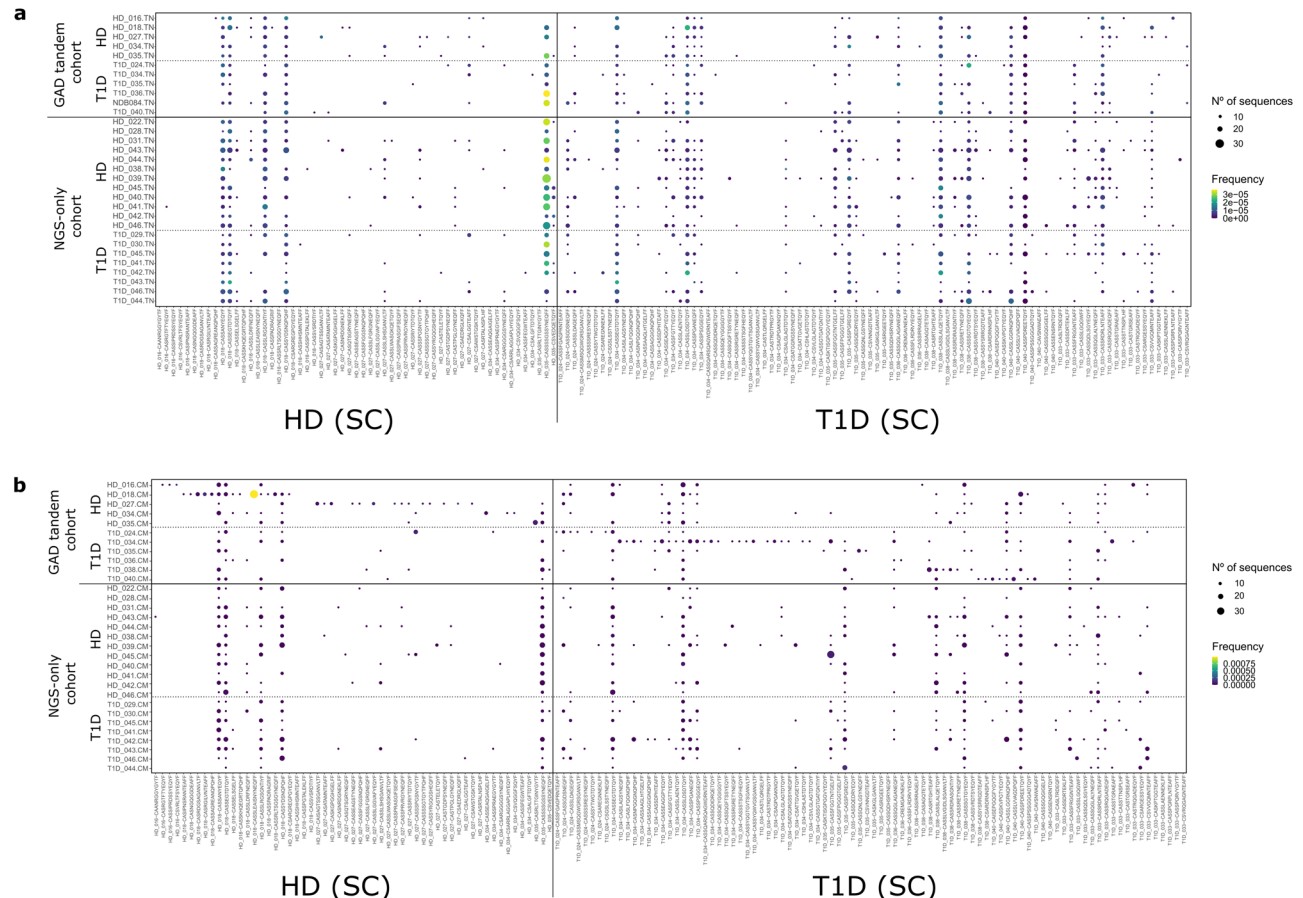

**Fig. 6 | GAD-specific amino acid clonotypes are public and convergent in TN and CM peripheral immune cell subsets.** We browsed the GAD-specific TCRB CDR3 amino acid clonotypes into 31 TN (**a**) and 31 CM (**b**) peripheral immune cell repertoires. We classified these repertoires into GAD tandem cohort (if both single-cell and deep sequencing took place) and NGS-only cohort (if we only performed deep sequencing). Colour represents the frequency of the tracked back clonotypes in the periphery, while dot size represents the numbers of unique TCRB CDR3 nucleotide sequences coding for each given amino acid clonotype (convergence).

decrease in the correlation value. This suggests that in CM subsets, frequency is less dependent on convergence- this is, the ease of generation does not seem to be the main variable that explains GAD and CMV clonotype presence in CM pools. For T1D-derived GAD clonotypes, the range of frequencies is comparable to those of HD-derived and CMV clonotypes; however, the convergence levels are much higher (compare red dots to blue and grey ones in Fig. 8g, h), indicating that the T cell clones expanded in CM in T1D were both easy to be generated in the thymus and prone to be expanded in the periphery, suggesting the possibility that convergent clonotypes in T1D are more easily expandable after antigen encounter.

When we look in detail at the frequencies of clonotypes in TN cells, we found that T1D-derived GAD clonotypes are equally frequent in HD and T1D patients, while HD-derived ones have lower frequencies in T1D TN cells (Fig. 9a, b, compare upper right and bottom left panels in heatmaps). This suggests that some physiological GAD clonotypes are rare in T1D. In general, GAD-specific clonotypes are more frequent in TN cells than CMV ones (Fig. 9a).

We next looked at frequencies in CM subsets: HD- and T1D-derived GAD clonotypes are of comparable frequencies within the same donor type (Fig. 9c, d), and these frequencies are comparable to those of CMV-specific clonotypes (Fig. 9c). Therefore, GAD clonotypes in CM are frequent, and not more frequent in T1D than in HD, suggesting that GAD clonotype frequency in the CM pool alone might not constitute a good biomarker of the disease.

In Treg subsets, HD-derived GAD clonotypes tend to have higher frequencies in HD Treg cells, which is not the case for T1D-derived ones, suggesting a much lower relevance of GAD clonotypes in T1D

Tregs than in their HD counterparts (Supplementary Fig. 16e, f). This is also the case for Tscm subsets (Supplementary Fig. 16g, h). Interestingly, CMV-specific clonotypes show low frequencies in Treg pools (Supplementary Fig. 16e), while their frequencies are much higher in Tscm cells (Supplementary Fig. 16g), highlighting the role of Tscm cells in the development of CMV-related immunity[62].

We next wondered whether specific GAD TCRB sequences were found in HD and separate ones in T1D patients. For this, we identified GAD clonotypes appearing in TN and CM peripheral repertoires from HD but not from T1D patients ("HD-only"), and GAD clonotypes appearing in peripheral repertoires from T1D patients but not from HD ("T1D-only"). In the case of TN repertoires, we found 11 HD-only and 6 T1D-only GAD clonotypes (Supplementary Data File 3). There were no differences in CDR3B length (14.18 *versus* 14 amino acids, respectively). Although low sample size constraints statistical analysis, V gene usage and clonotype distribution among genex clusters were different between HD-only and T1D-only clonotypes (Supplementary Fig. 17a, b and Fig. 9e).

In the case of CM repertoires, we found 34 HD-only and 33 T1D-only GAD clonotypes (Supplementary Data File 4). There were no differences in CDR3B length (14.4 *versus* 14.5 amino acids, respectively). Regarding V-gene usage, *TRBV20-1* and *TRBV28* are enriched in HD-only, and *TRBV29* and *TRBV6-5* enriched in T1D-only GAD clonotypes (Supplementary Fig. 17c). T1D-only GAD clonotypes present different amino acid usages in almost all CDR3B positions when compared to HD-only ones, with a preference for hydrophobic amino acids in positions P6 and P7 in the case of 13 amino acid CDR3B's (Supplementary Fig. 17e, f).

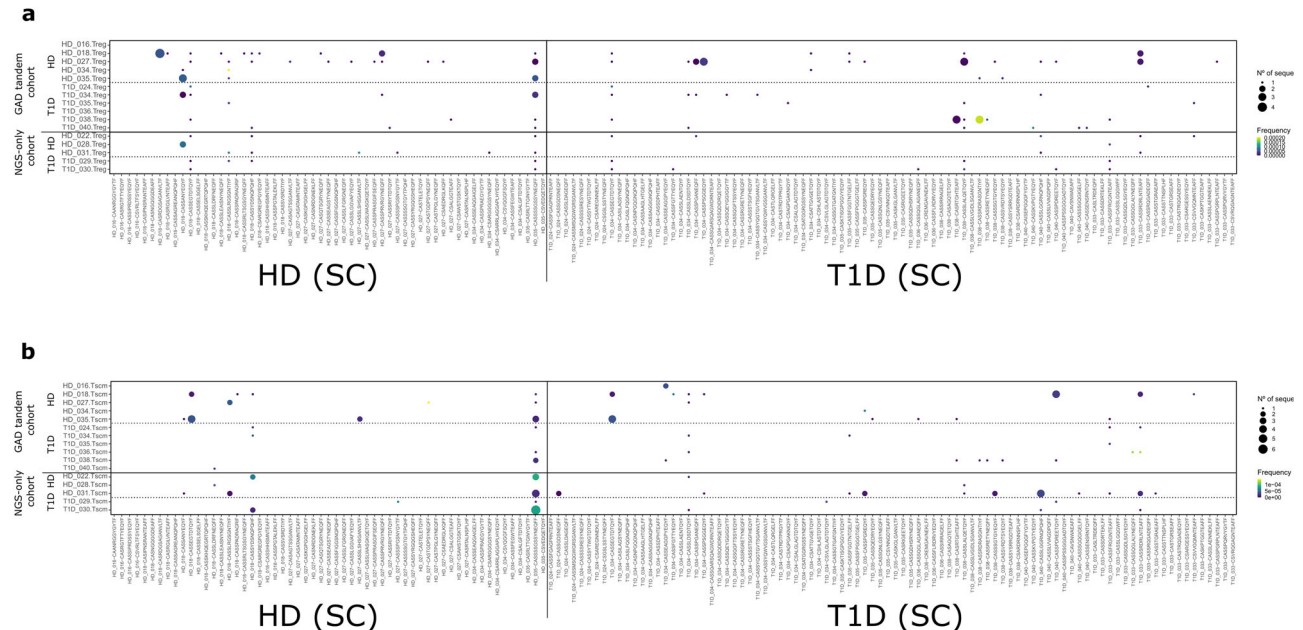

**Fig. 7 | Some GAD-specific amino acid clonotypes are public in Treg and Tscm peripheral immune cell subsets.** We browsed the GAD-specific TCRB CDR3 amino acid clonotypes into 16 Treg (**a**) and 16 Tscm (**b**) peripheral immune cell repertoires. We classified these repertoires into GAD tandem cohort (if both single-cell and deep sequencing took place) and NGS-only cohort (if we only performed deep sequencing). Colour represents frequency of the tracked back clonotypes in the periphery, while dot size represents numbers of unique TCRB CDR3 nucleotide sequences coding for each given amino acid clonotype (convergence).

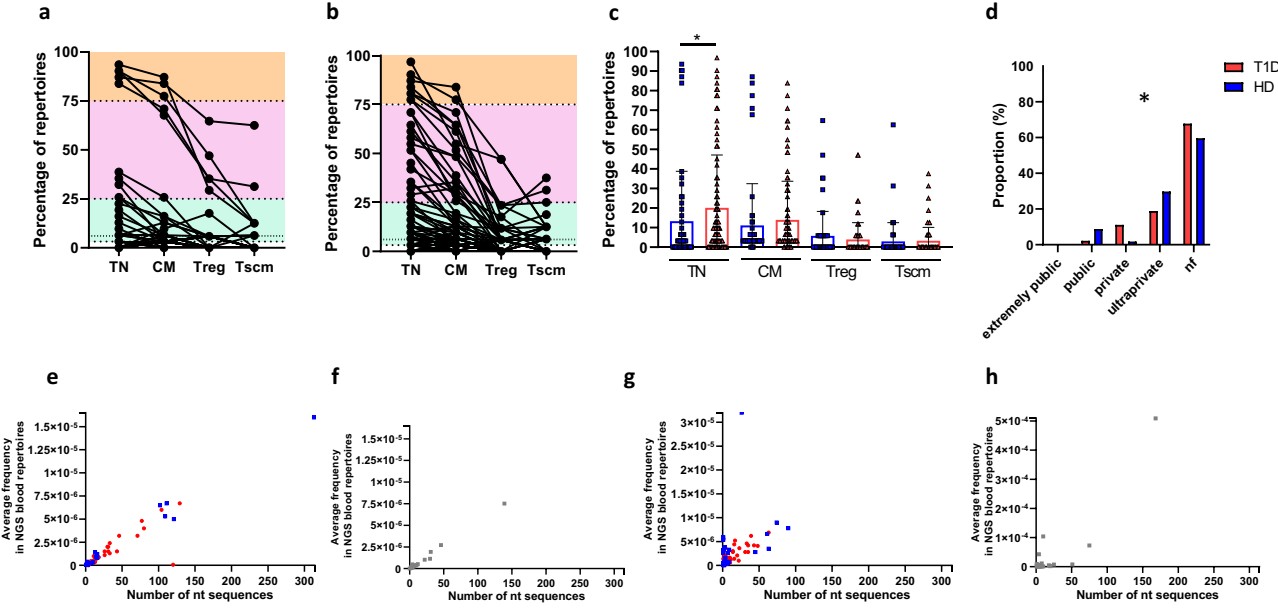

**Fig. 8 | GAD-specific TCRB CDR3 amino acid clonotypes from T1D patients show differential publicity and convergence features within peripheral immune repertoires.** **a**, **b**: Percentage of peripheral immune repertoires where each GAD-specific TCRB CDR3 amino acid clonotype is tracked into, disaggregated per type of subject (**a** HD, and **b** T1D). Orange shade: extremely public clonotypes (present in ≥ 75% of individuals). Pink shade: public clonotypes (present in 25.0%–74.9% of individuals). Green shade: private clonotypes (present in 3.23%–24.9% of individuals) or ultraprivate (only found in the same single-cell donor, 3.23%). For Treg and Tscm, the threshold for "ultraprivate" is 5.88 and 6.25 respectively (faint dotted lines). **c** Percentage of repertoires where GAD-specific TCRB CDR3 amino acid clonotypes are tracked back into (HD: 32 clonotypes into TN, 51 into CM, 23 into Treg and 10 into Tscm. T1D: 66 clonotypes into TN, 77 into CM, 29 into Treg and 25 into Tscm. Two-sided Mann-Whitney U test. *$p = 0.013$). Blue: HD. Red: T1D. **d** Proportions of each publicity-based clonotype category in Treg repertoires (two-sided Chi-square test. * $p = 0.0025$. Blue: HD. Red: T1D). **e–h** correlations between number of convergent nucleotide sequences and frequency for each GAD-specific TCRB CDR3 amino acid clonotype tracked back into peripheral immune repertoires (Spearman correlation). **e**, **f** correlations for GAD (**e**) and CMV (**f**) clonotypes found in TN subsets. **g**, **h** correlations for GAD (**g**) and CMV (**h**) clonotypes found in CM subsets. **e**, **g** Blue: HD-derived GAD clonotypes. Red: T1D-derived GAD clonotypes. Error bars represent standard deviations.

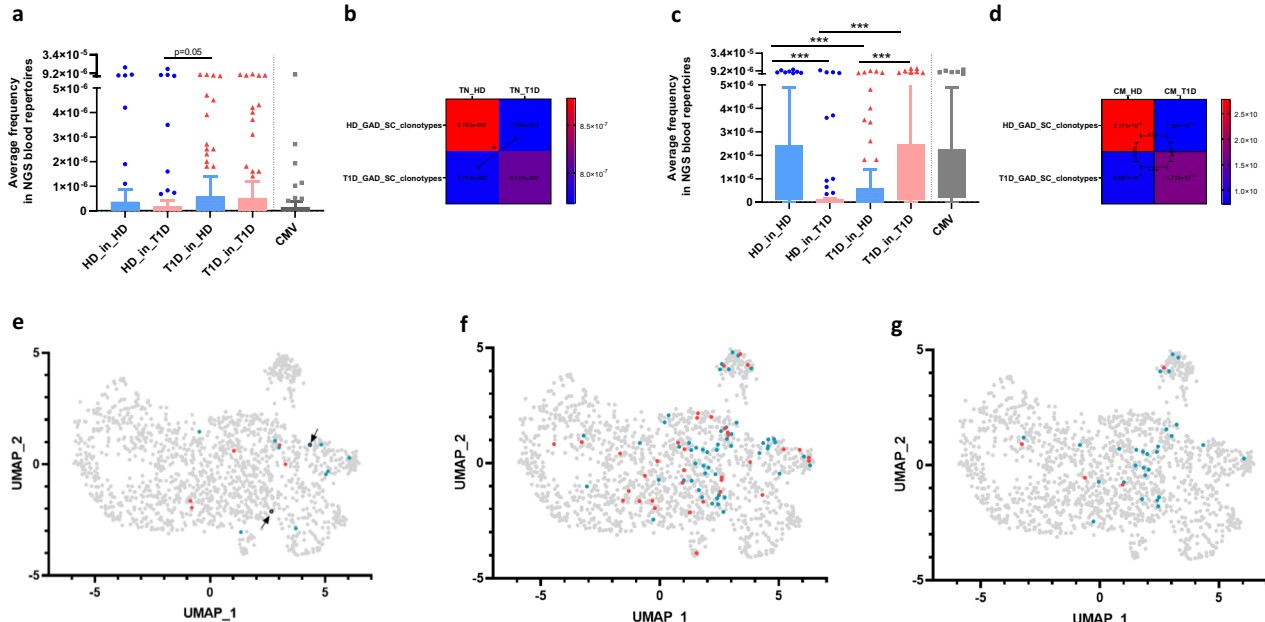

**Fig. 9 | Frequency, clonal sizes and phenotypes are different between HD- and T1D-derived GAD clonotypes. a, c** Frequency of GAD-specific TCRB CDR3 amino acid clonotypes in TN (**a**) and CM (**c**) peripheral immune repertoires, disaggregated by type of donor (57 clonotypes for HD, 90 for T1D and 64 for CMV). Shown are Tukey boxplots: boxes are interquartile range (25% to 75%), whisker (up) is 75th percentile plus 1.5 times the interquartile range (IQR). The centre is the 50th percentile. Individual dots are values that are greater than the whisker. Two-sided Mann Whitney U test with Bonferroni correction (***: $p < 0.001$). The frequency of each clonotype was calculated as the average frequency for all nucleotide sequences coding for the same amino acid sequence (See "Methods" section). **b, d** heatmaps showing the means of frequencies, and the corresponding $p$-values (two sided Mann Whitney U test with Bonferroni correction), of data shown in **a**, **c**. **e, f** we identified GAD clonotypes appearing in peripheral repertoires from HD but not from T1D patients ("HD-only", in blue), and GAD clonotypes appearing in peripheral repertoires from T1D patients but not from HD ("T1D-only", in red), and represented them on the UMAP space. GAD clonotypes found in TN repertoires are shown in (**e**), while those found in CM ones are shown in (**f**). Arrows in (**e**) depict expanded clonotypes. **g** HD-only (blue) and T1D-only (red) GAD clonotypes tracked into CM subsets which show clonal expansions.

Interestingly, we found that many HD-only GAD clonotypes are located in the regulatory/multiple TF cluster #5, while T1D-only GAD clonotypes are more often found in the TGF-β+ cluster #3 (Supplementary Fig. 17d and Fig. 9f). We also observed that 29% (10/34) of HD-only clonotypes were expanded, this is, were found in more than two cells with the same TCRB and TCRA (Supplementary Data File 4). Importantly, 60% on these expanded clonotypes have at least one cell located in the regulatory/multiple TF cluster #5 (Fig. 9g), suggesting that these expanded clones include both effector and regulatory phenotypes. In T1D-only GAD clonotypes, however, there are almost no expansions (6% (2/33)), and in none of these two cases there are any cells located in cluster #5 (arrows in Fig. 9e). This is suggestive of HD peripheral repertoires being populated by some exclusive GAD-specific clones of higher sizes, which include potential regulatory phenotypes, and that are missing in T1D patients.

In summary, we found that T1D-derived GAD clonotypes are more often of public nature and convergent, with a peripheral frequency comparable to that of HD-derived GAD clonotypes, and that this convergence is maintained even after clone expansion in CM subsets. Interestingly, patient-derived GAD-specific clonotypes are frequent in TN cells from HD, while HD-derived GAD-specific clonotypes are rarer in T1D TN cells. GAD clonotypes are frequent in CM pools, both in HD and T1D, and HD-only and T1D-only GAD clonotypes present different features, including the lack of regulation-related TCRB clonotypes in patients.

## Discussion

Here we show that autoreactivity against GAD is the norm, and not the exception, pointing towards the existence of physiological and regulated autoimmunity against GAD which, in T1D, combines with disease-specific autoimmune features. We observed, through ELISPOT and

AIM assays, that many HD were highly responsive against GAD. We also identified memory IL-2neg cells expressing cytokines as well as activated TN cells, suggesting that our AIM assay captures not only proliferative phenotypes, but also non-proliferative ones.

When we performed single-cell level approaches, we found that GAD-specific responses include a plethora of phenotypes, including easy-to-categorise subsets such as Th1, Th2 or IL-22+, but also Treg-like with or without transcription factor heterogeneity, memory Tfh, quiescent TGF-β+ cells and hyporesponsive ones. The existence of such a variety of responses highlights the importance of performing multilayered and orthogonal approaches when dissecting immune responses against autoantigens.

Some of these phenotypes are physiological as are similar in HD and T1D patients, including highly activated Th2 cells. We and others have previously described IL-13 expression/secretion against GAD, both in clinical trials after GAD administration or in ex vivo and in vitro analyses[27,28]. In particular, we found that GAD-specific IL-13 responses in the TrialNet TN08 study [https://repository.niddk.nih.gov/] are present at baseline and massively increased after GAD administration[28]. Th2 expansion was also observed in T1D patients during remission phases[63]. We now expand this information, showing that these GAD-specific, IL-13 responses are also found in non-immunised healthy individuals, suggesting that Th2-like responses against GAD constitute a physiological response.

We unexpectedly observed Th1 responses against GAD in HD; interestingly, it was recently reported that insulin-specific CD4+ T cells in HD also present a Th1 phenotype[37]. We previously described the presence of IL-22+ CD4+ T cells specific for insulin epitopes and neoepitopes, identified through the AIM assay[64], similarly to what we report here for GAD. Besides, in that study, IL-10 was undetectable through the AIM assay[64]; we now find few cells expressing *IL-10*, possibly due

the short time of culture (16 h) not being enough to generate detectable levels of *IL-10* mRNA. However, the transcription factor c-Maf is the master regulator of IL-10[59,60], and we find that *MAF* expression is significantly higher in regulatory cluster #5.

We previously reported that naïve beta cell antigen-responsive CD4[+] T cells are present in neonates[65] and identified a divergent naïve CD4[+] T cell response to beta cell antigens that preceded the appearance of beta cell antigen-responsive memory T cells and beta cell autoantibodies[66]. Now we extend this finding focusing on recently activated CD4[+] T cells instead of just proliferating ones: we observed that there is a significant enrichment for GAD-specific true naive cells in patients compared to healthy donors. Therefore, our GAD-specific naïve cells potentially represent pre-primed, partially committed T cells, reflecting a state of latent autoreactivity. Other studies also observed, in overall CD4[+] cells, an increase of naïve cells in newly diagnosed patients[37].

We previously showed that thymic events are altered in T1D[40], and we also reported that some T cells with the potential to recognise GAD epitopes in GAD-immunised patients are expanded from a public TCRB repertoire[28]. In the last study, however, antigen-specific clonotypes and peripheral repertoires were not from the same donors[28]. Now, we track back GAD-specific clonotypes into peripheral TCR repertoires isolated from the same donors and blood sample, increasing the significance and relevance of tracking back TCRs of interest. To our knowledge, only us in our previous study[41] performed tandem TCR sequencing in T1D, which now we accomplish through more advanced sequencing approaches, and complement with index sort and single gene expression profiling. This tandem sequencing and tracking back allows for the precise quantification of frequencies of antigen-specific TCRB nucleotide and amino acid sequences in blood.

The chances of finding the same TCRB nucleotide sequences in two unrelated individuals is low, due to the random nature of thymic VDJ rearrangement[67]. As expected, most of the GAD-specific TCRB CDR3 nucleotide sequences, when found in CM, Treg and/or Tscm subsets, were only present in the same donor. However, we found unexpected publicity of GAD TCRB CDR3 nucleotide sequences in TN pools, while this was not the case for a viral antigen. This is suggestive of GAD sequences not only being easier to generate during recombination, but also being purposely selected in the thymus. Given the extensive remodelling of pancreas and brain, the only two GAD-expressing organs, during neonatal phases[2–5], it is it is tempting to speculate that certain GAD clonotypes might be favoured during thymic selection, to support a physiological autoimmune process needed during remodelling.

We then undertook a systematic track back of GAD-specific clonotypes at the amino acid level: this allows us to interrogate the convergence levels of these clonotypes and their functional relevance, determined as peripheral frequency. We observed an increase in convergence of T1D-derived GAD clonotypes found in CM subsets, which cannot be explained by thymic ease of generation alone, suggesting that convergent clonotypes are more prone to be expanded in T1D patients. We unexpectedly observed that T1D-derived GAD clonotypes were found in TN cells from HD with the same peripheral frequencies as in patients. However, HD-derived GAD clonotypes are rarer in TN cells from patients, are more expanded and are more often related to regulatory phenotypes, maybe indicating a potentially protective role of these clonotypes. When GAD-specific clonotypes are found in CM pools, we observed that these clonotypes have comparable high frequencies in HD and T1D patients, with low sharing between the two groups of individuals. This suggests that these GAD repertoires in CM might be driven by different biological processes, namely physiological autoimmunity in HD, and diabetogenic autoimmunity in T1D. These frequencies are comparable to CMV ones.

These results suggest that the thymus is poised towards GAD-specific clonotype selection, as noted by the high frequency of GAD

clonotypes in peripheral conventional TN cells, and that biological forces taking place after selection and antigen encounter lead to specific health and patient-related expansions of equal sizes in CM, Treg and Tscm peripheral populations. Importantly, GAD clonotypes seem to be more relevant than CMV ones in peripheral Treg cells, suggesting that regulation of responses against GAD is more needed than for CMV.

We are aware that peripheral blood only partially reflects the actual nature of immune responses in T1D. However, this is the primary sample employed in T1D clinical trials for readouts, given its ease of access and the ethics involved in sampling the pancreas in live patients. Additionally, other studies have shown that circulating antigen specific CD4[+] T cells are reliable indicators of islet autoimmunity[37]. It is also important to note that we focused on the expression of a panel of immune-related genes, given our previous expertise in this technology[28,68,69]; a wider analysis including more genes could show up/down regulation of other genes in these clusters. Equally, only TCRB chains were sequenced in the peripheral immune subsets. Current technologies are not fully available yet for pairing TCRA and TCRB reads by next generation sequencing. Future technical advances will allow this, propelling new studies on paired repertoires in bulk populations.

In summary, we propose a model whereby protective autoimmunity against GAD is needed physiologically, due to the intense neonatal remodelling of the pancreas (Fig. 10). During foetal life, the beta cell mass expands due to beta cell neogenesis and, although it continues to expand in neonates, the net beta cell mass remains unchanged due to an accompanying wave of apoptosis occurring at the time of birth[4]. After remodelling and cell death, GAD-specific T cells selected by the thymus lead the clearance of the area and the repair of the tissue. The presence of peripheral regulatory mechanisms in health ensures that autoreactivity against GAD is maintained under control throughout life, as further insults such as nerve injury or viral infections could potentially release GAD to the circulation. In T1D, some of these physiological phenotypes also exist, while others are altered, propelling the uncontrolled destruction of beta cells. Further alterations underlying T1D, including thymic defects leading to excessive TCR diversity and shorter length[40], defects in peripheral regulatory mechanisms[15,70,71] and altered responses against other autoantigens (e.g (pre)(pro)insulin, IA-2, ZnT8, neoantigens)[15,72], combine to surpass the physiological autoimmunity, leading to overt T1D. Further studies in bigger cohorts, including children diagnosed with T1D, are needed to fully dissect the nature of physiological *versus* diabetogenic autoimmunity, helping to inform immunotherapy clinical trials.

## Methods
### Human donors and study design
Blood was obtained from 46 patients with new onset type 1 diabetes (less than one year since diagnosis; average 3.8 months) and 46 healthy donors and assays ran as summarised in Supplementary Fig. 1. The study was approved by the National Research Ethics Service, NRES Committee London-Bromley, REC reference 08/H0805/14. All uses of human material have been approved, and all recruited volunteers provided written informed consent. Patients were compensated for their time. Donors were typed for *DRB1\** alleles by the Tissue Typing Service at Guy's Hospital (London, UK). Demographic data, HLA and autoantibody status are shown in Supplementary Table 1. PBMCs were immediately isolated by density gradient centrifugation from heparinised Vacuette tubes (Greiner Bio-one, Gloucestershire UK) using Lymphoprep (Axis-Shield PoC AS, Oslo, Norway) as indicated by the manufacturer, and used always fresh in all the assays indicated below. We kept the same percentage of *DRB1\*0301* (DR3) and *DRB1\*0401* (DR4) individuals, as well as similar male/female ratios, in each sub study described below. By doing this we control for HLA and sex

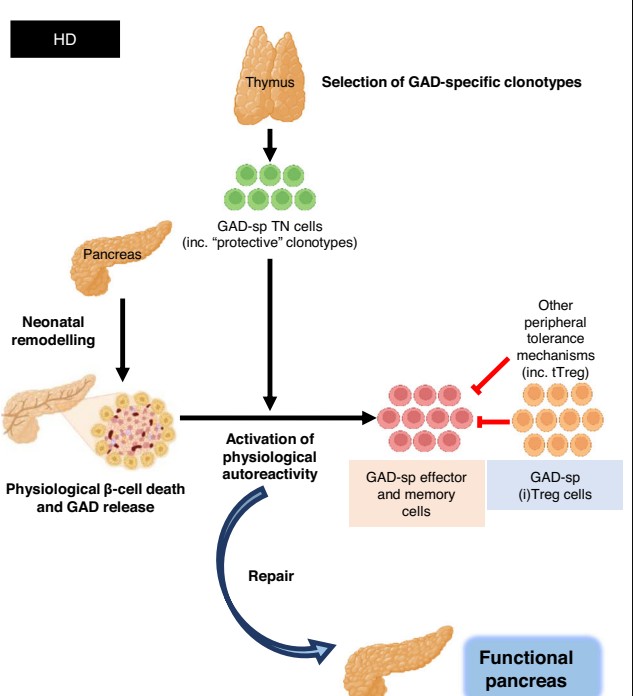
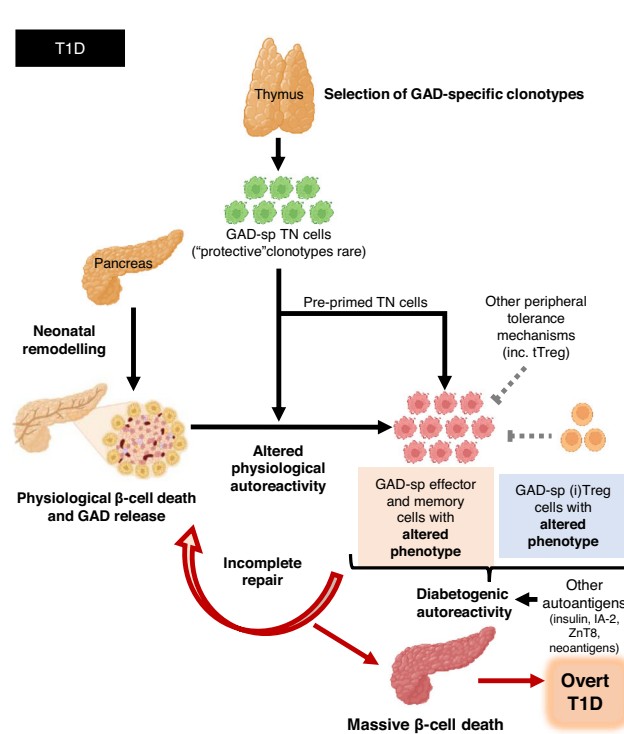

**Fig. 10 | Model of physiological and diabetogenic autoreactivity against GAD.**
In HD (left), GAD-specific clonotypes are selected in the thymus, generating GAD-specific TN cells which seem to include "protective" clonotypes. During neonatal remodelling of the pancreas, physiological beta-cell death occurs, releasing GAD. GAD-specific TN cells get activated by antigen in this context of cell death, initiating a repair mechanism, leading to a fully functional pancreas. During this process, GAD-specific effector and memory cells are generated, presenting phenotype features predisposed to tissue remodelling and repair. Equally, peripherally induced GAD-specific Treg (iTreg) cells are generated and, in potential combination with thymic derived Treg (tTreg) cells and other peripheral tolerance mechanisms, keep the memory GAD cells under control. In T1D patients (right), GAD-specific clonotypes are also selected in the thymus- however, alterations during recombination events generate a repertoire of GAD-specific TN cells that is more public.

Additionally, "protective" clonotypes are rarer. During pancreas remodelling and GAD release, an altered physiological autoreactive process occurs: the GAD-specific effector and memory cells generated during the process are of altered phenotypes which are not sufficient to drive appropriate remodelling and repair mechanisms. Additionally, effector and memory cell generation is further fuelled by pre-primed TN cells. As such, beta-cell death continues. Less induced Treg cells are generated and, if so, their phenotypic features do not allow them to contain the effector responses. Thymic Tregs and other peripheral tolerance mechanisms, previously shown to be altered in T1D, cannot contain this diabetogenic autoreactivity. This combination of altered physiological and diabetogenic autoreactivity converge to propel beta cell death, leading to overt T1D. GAD-sp: GAD-specific. Some icons have been created in BioRender. Gomez-tourino, I. (2024) BioRender.com/q46s625.

biases, while allowing for a wide and agnostic analysis of responses against any naturally processed and presented GAD epitopes.

## Enzyme-linked immunospot (ELISPOT) assays

Interferon-γ (IFN-γ) and IL-10 production was detected by ELISPOT assay[64] in 40 patients and 36 age/sex/HLA matched healthy donors (Supplementary Fig. 1 and Supplementary Data File 1). Briefly, fresh PBMCs supplemented with whole GAD or GAD diluent (Dyamid) were dispensed into 48-well plates at a density of $2 \times 10^6$ cells/well in 0.5 ml RPMI-1640 supplemented with antibiotics (TC medium; Life Technologies Ltd.) and 10% human AB serum (Sigma, Dorset, United Kingdom). Pediacel, a penta-vaccine, was obtained from Sanofi Pasteur Ltd. (Guildford, U.K.) and used at 1 μl/ml as positive control. Pre-warmed TC medium/10% AB serum was added 24 h later, and 48 h after stimulation cells were washed and resuspended in TC medium containing 10% human AB serum and brought to a concentration of $10^6/300$ μL; 100 μl was dispensed in triplicate into wells of 96-well ELISA plates. (Nunc Maxisorp; Merck Ltd., Poole, United Kingdom) preblocked with 1% BSA in PBS and precoated with monoclonal anti-IFN-γ or anti-IL-10 capture antibody (U-Cytech, Utrecht, The Netherlands). After capture at 37 °C for 20–22 h, plates were washed in PBS/Tween 20, and spots developed according to the manufacturer's instructions. Plates were dried, and spots of 80–120 mm were counted in a BioReader 3000 (BioSys, Karben, Germany), and data were expressed as

stimulation index (SI) values (mean spot number of GAD/mean spot number of GAD diluent). SI values ≥ 3 was taken to indicate a positive response.

## Activation-induced marker assay

The activation-induced marker (AIM) assay was performed for 10 patients and 11 age/sex/HLA matched healthy donors (Supplementary Fig. 1 and Supplementary Data File 1) using an assay previously developed[42–44], and adapted by us[41] for this purpose. Briefly, fresh PBMCs ($2 \times 10^7$ per antigen condition) were incubated at $10^6$ cells/ mL in 48-well plates (37 °C, 5% $CO_2$, 1 mL/well) in RPMI 1640 with Glutamax supplemented with penicillin/streptomycin, Amphotericin B (Fisher Scientific, 15140 and 15290) and 10% human AB serum (Sigma, H4522) containing 2 μg/mL of anti-CD40 monoclonal antibody (Biolegend, clone G28.5, 303611) and 10 μg/mL of either GAD protein or GAD diluent, or CMV grade 2 antigen (containing antigens from all parts of the virus cycle of replication, Microbix Biosystems Inc, EL-01-02). As a positive control $2 \times 10^6$ PBMCs were incubated as above with 10 μg/mL of the polyclonal stimulus Staphylococcal enterotoxin B (SEB, Sigma-Aldrich, S4881). The negative control consisted of $2 \times 10^6$ PBMCs incubated as above with no additional stimulus. Non-adherent cells were harvested after 18 h, washed, and stained with the following fluorochrome conjugated monoclonal antibodies (indicated in brackets is clone name, catalogue number and volume in μL used to stain 2 ×

$10^6$ PBMCs): anti-CD14 (TuK4, MHCD1428, 2 µL), anti- CD19 (SJ25-C1, MHCD1928, 2 µL) (Invitrogen); anti-CD3 (SK7, 641415, 2 µL), anti-CD154 (TRAP1, 555700, 2.5 µL), anti-CD69 (FN50, 555530, 2.5 µL), anti-CD45-RO (UCHL1, 337168, 2.4 µL), anti-CD95 (DX2, 561978, 3 µL) (all from BD Biosciences); anti-CD4 (SK3, 46-0047-42, 3 µL. eBiosciences); anti-CD27 (O323, 302830, 1.2 µL. Biolegend) and LIVE/DEAD Fixable ViVid Dead Cell Stain (Molecular Probes, L34955). After staining, cells were washed and acquired in a BD FACSAria II flow cytometer or a BD FACSCanto II. The gating strategy is shown in Supplementary Fig. 3. For surface phenotype analysis, we normalised the frequency of each phenotype in each individual (to take into account differences in number of non-gated events).

### Single cell index sort and analysis

We single cell index sorted GAD-specific $CD4^+$ T cells from 7 T1D patients and 5 age/sex/HLA matched HD, and CMV-specific $CD4^+$ T cells from 2 HD and one T1D patient (Supplementary Fig. 1 and Supplementary Data File 1). Cells were stained as indicated above. Flow cytometric sorting took place in a BD FACSAria II flow cytometer (Becton Dickinson, San Jose CA, USA) with FACSDiva software version 7.0 and with the index sorting option activated; this allows for all fluorescence values to be recorded independently for each cell sorted. The fluids system was flushed before sorting with RNAse Zap (Thermofisher). The gating strategy is shown in Supplementary Fig. 3. Live $CD3^+$ $CD4^+$ $CD154^+$ $CD69^+$ cells were index sorted with a flow rate of 2000-5000 events/s into 96-well PCR plates (Thermofisher) containing 5 µL of RNAse/DNase-free PBS, using a 70 µM nozzle. Plates were snap frozen in dry ice and stored at -80C until processing.

The cell surface phenotype of each individual $CD154^+$ $CD69^+$ cell sorted was assigned post hoc by the expression of CD45R0, CD27 and CD95; naïve/memory gates were drawn using all $CD4^+$ events, and then pasted into the $CD154^+$ $CD69^+$ events, as follows: true naïve (TN: $CD45R0^{neg}$, $CD27^+$, $CD95^{neg}$), central memory (CM: $CD45R0^+$, $CD27^+$), effector memory (EM: $CD45R0^+$, $CD27^{neg}$) non-terminated effector memory (NTEM: $CD45R0^{neg}$, $CD27^{neg}$), or stem cell-like memory (Tscm: $CD45R0^{neg}$ $CD27^+$ $CD95^+$) (Supplementary Fig. 3).

### Cell line generation and proliferation assays

Bulk live $CD3^+$ $CD4^+$ $CD154^+$ $CD69^+$ cells were sorted into U-bottom 96-well plates containing irradiated fresh mixed feeders and PHA and fed every other day with a final concentration of 10% Cellkine (Zeptometrix). Cells were restimulated after 12-14 days, and Cellkine was subsequently reduced. Proliferation assays were performed as previously described[73]. All conditions were run in triplicate and proliferation readings (CPM) averaged.

### Single-cell gene expression analysis

Single-cell PCR was performed as previously described[66,74]. Briefly, cDNA was synthesised directly from single cells using qScript cDNA Supermix (Quanta Biosciences). A first round of PCR was done on all cDNA using TATAA GrandMaster Mix (TATAA Biocenter, Göteborg, Sweden) in the presence of a mix of 31 tagged (HTSP-"tag") variable region primers covering all subgroups of TCR Vα and Vβ genes, an alpha constant region primer, a beta constant region primer and 96 primers for preamplification of 48 genes of interest, including cytokines, transcription factors, chemokines and lineage markers (see Supplementary Data File 5 for oligo sequences). A second round PCR was performed separately for TCRA and TCRB on 3 µl of the first round product, using the HTSP-"tag"-primer and a nested constant region primer (for alpha or beta, depending on the PCR reaction), with Takara ExTaq (Takara BIO INC) (Supplementary Data File 5). The PCR products were Sanger sequenced using the HTSP-tag primer. The sequencing results were referenced to the IMGT database[75], the CDR3A and CDR3B sequences extracted and compared across individual cells using KNIME 4.5.1[76]. To identify similar TCRB CDR3 sequences we employed

GLIPH[61] version 1.0 with default parameters, except for simdepth 10000 and naïve CD4 cells from the files distributed with the software (https://github.com/immunoengineer/gliph).

### Microfluidics single-cell quantitative PCR and data analysis

To quantify gene expression on single cells, real-time PCR was performed on 2.7 µL of exonuclease-digested $1^{st}$ PCR product in the Bio-Mark™ HD System (Fluidigm Corporation, South San Francisco, CA), using the 96.96 Dynamic Array IFC according to the GE 96×96 Fast PCR + Melt protocol with SsoFast EvaGreen Supermix containing Low ROX (Bio-Rad, Hercules, CA) and 5 µM of primers in each assay[77] (Supplementary Data File 5). The following target genes were analysed: *IL4, Tbet, IL17A, RORA, CD40, IL13, TGF-β, CCR3, CXCR5, CCR10, CD52, INF-γ, TNF-α, GATA3, IL9, Bcl6, RANTES, IL17F, ICOS, IKZF2, IL22, CD4, REL, IL18RAP, CTLA4, IL10, RGS16, IL2, AHR, MAF, CCR6, Egr2, EOMES, CD8, GMCSF, PD1, CCR7, IL21, CD3e, CCR4, GITR, RORC, CCR5, FOXP3, NFATC2, CD127, CD134,* and *SRP14* (as a housekeeping gene). Raw data was analysed using Fluidigm Real-Time PCR analysis software. Melting curves were visually inspected and any reaction that produced incorrect products were removed manually. Positive and negative controls for each of the steps (RT-PCR, preamplification, and qPCR) were run, together with a plate calibrator consisting of a titrated amount of exonucleased preamplified SRP14 product. The minimum Ct for each cell and gene was selected from the duplicates, missing values replaced by 32, data transformed so the highest Ct becomes 0, and cells with 0 expression in all genes removed. Downstream data analysis was done using KNIME version 4.5.1[76], Seurat package version 3.1.3, MAST package version 1.12 [78], and in-house R scripts. Normalised and batch-corrected data was scaled for dimensionality reduction. UMAP was based on 10 PCs with defaults Seurat's options. Clusters were based on shared nearest neighbour graph and Louvain algorithm, with Resolution=1.

Next, we ran statistical tests to infer marker genes defining each cluster. For that, we employed the hurdle model, as implemented in MAST[78], to identify differences among clusters: this test provides three $p$-values, for (i) continuous differences (genes with higher or lower than average expression), (ii) discontinuous differences (genes expressed in more or less cells than average) and (iii) an overall $p$-value. We corrected for multiple comparisons (Bonferroni), and only kept those genes where the average logarithm of the fold change was higher or lower than 0.25.

For the analysis of distribution of HD and T1D cells in the UMAP space we computed, for each individual, what fraction of cells are located in a given cluster. Then, for each cluster, we run a Wilcoxon test for HD versus T1D and applied the multiple comparisons Bonferroni correction.

### Track back of antigen-specific clonotypes into peripheral immune repertoires

We previously sorted and performed deep sequencing of TCRB CDR3's of TN, CM, Treg and Tscm peripheral cell subsets from 14 patients and 17 age/sex/HLA matched HD[40]: a mean of $94 × 10^6$ cells per donor were directly stained for sorting and deep sequencing of these TCRB CDR3, while the remainder PBMCs were used to set up the ELISPOT, AIM assays, index sort, single cell TCR PCR and qPCR as described above. GAD and CMV-specific TCRB nucleotide and amino acid sequences were searched into the peripheral subsets using in-house R scripts. As several nucleotide sequences can codify for the same amino acid clonotype, we calculated the average frequency of all nucleotide sequences coding for that particular amino acid clonotype to infer the frequency of each amino acid clonotype. This was only needed when tracking back at the amino acid level.

For publicity determination at the amino acid level, we classified the clonotypes as extremely public (present in ≥ 75% of individuals), public (present in 25.0%–74.9% of individuals), private (present in

3.23%–24.9% of individuals) or ultraprivate (only found in the same single-cell donor, 3.23%). For Treg and Tscm, the threshold for "ultraprivate" is 5.88 and 6.25 respectively.

For the analysis of HD-only and T1D-only GAD clonotypes we identified, separately for TN and CM subsets, those clonotypes not being tracked back into any T1D patient ("HD-only") or into any HD ("T1D-only"). For the genex distribution determination we employed all tracked back clonotypes, including those appearing in more than one cell (expanded). CDR3B length, and V gene and amino acid usage were analysed for unique clonotypes, to avoid biases due to certain clonotypes coming from more than one cell. Amino acid logos were generated using pLogo[79] with default settings.

### In vitro TCR expression and stimulation

Expression of recombinant TCR in the 5KC reporter system was performed as previously described[80]. To construct TCR- expressing vectors, a fragment containing a human T cell receptor alpha variable gene (TRAV) variable segment, one with the mouse TRAC*01 gene fused to the viral P2A sequence and one with a human T cell receptor beta chain variable segment (TRBV) were synthesised (Genewiz, Azenta) with overlaps and joined to the backbone of an Murine Stem Cell Virus (MSCV)- based vector containing a murine TRAV signal peptide and the murine TRBC1 gene by Gibson assembly[81]. Vectors were introduced into 5KC hybridoma cells containing the 8xNFAT-ZsG reporter, muthCD4 along with either LSSmOrange (LO), blue fluorescent protein (BFP) or E2 Crimson (CR) or combinations thereof, by transfection into Phoenix cells and subsequent transduction. All cell lines were generous gifts from Professor Maki Nakayama and have been described[82]. CD3$^+$ (TCR expressing) live cells were FACS-sorted (ARIAIII$^{TM}$, BD) and further expanded in culture medium. For antigen stimulation, 150,000 APCs (the CD3$^-$, CD4$^-$, CD8$^-$ live cell fraction of PBMCs) were incubated with 75ug/ml GAD (T cell-GAD, 10-4508029-01, Dyamid) or, as positive and negative controls, 2.5ug/ml hamster NALE anti-mouse CD3e (clonotype 145 2C11) or Dyamid Control, respectively, for 4 hours. 20,000 recombinant TCR- expressing 5KC cells were then added for overnight incubation in a total volume of 200uL in culture medium. Draq7$^{TM}$ (BioStatus) was then added and the expression of the ZsG- reporter was acquired on a LSRFortessa$^{TM}$ (BD) and analysed with FlowJo$^{TM}$ v10.8.

### Statistical analysis and graphical representation

For comparisons of two groups, paired or unpaired two-tailed student t-test (for normally distributed variables) or Mann Whitney U-test (for other variables) were used. For comparison of more than two groups, ANOVA + Tukey (for normally distributed variables) or Kruskal Wallis + Dunn (for other variables) were used. Correlations were calculated using Pearson or Spearman test. When pertinent, Bonferroni multiple comparison correction was performed. Chi-square or Fisher's exact test was used for comparing groups in analysis involving qualitative variables. Unless otherwise stated, bars represent means and error bars standard deviations. In Tukey boxplots, the boxes are interquartile range (25%–75%), whisker (up) is 75th percentile plus 1.5 times IQR. The software used for data analysis and graphical representation of the data included SPSS (IBM), GraphPad Prism 8, in-house R scripts, and KNIME 4.5.1[76].

### Reporting summary

Further information on research design is available in the Nature Portfolio Reporting Summary linked to this article.

## Data availability

TCRB CDR3 sequencing data of peripheral immune cell subsets have been deposited and made public in the Open Science Framework database (DOI 10.17605/OSF.IO/YDGTV)70: http://osf.io/ydgtv/), in GEO (GSE272431), and published in reference[40]. GAD- and CMV-specific clonotype sequences have been deposited in GenBank (IDs PP952812-PP953496). The TCRB CDR3 sequencing data, along with associated study metadata, for both the peripheral immune cell subsets and the GAD- and CMV-specific clonotypes are stored in the AIRR Data Commons and can be searched and downloaded using the iReceptor Gateway[83,84] (https://gateway.ireceptor.org) study IDs "DOI:10.21417/B7C88S" and "IR-T1D-000003" respectively. Source data are provided with this paper. All other data are available in the Supplementary Data files. Correspondence and requests for materials should be addressed to I.G.-T. Source data are provided with this paper.

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

## Acknowledgements

The study was supported by the Xunta de Galicia (ED431C 2023/28, to I.G.-T.) and the National Institute for Health Research (NIHR) Biomedical Research Centre based at Guy's and St Thomas' National Health Service (NHS) Foundation Trust and King's College London (IS-BRC-1215-20006, to M.P). I.G.-T. was also initially supported by a Marie Curie Intra-European Fellowship (PIEF-GA-2012-327908). M.K. is supported by a predoctoral fellowship from Xunta de Galicia (ED481A 2022/422). We are grateful to study volunteers for their participation, to Laura Eckhardt and Dr Jake Powrie for assistance in participants recruitment, to Annett Lindner for cloning, and to Sefina Arif, Ruben Varela-Calvino and Martin Eichmann for critical review of the manuscript.

## Author contributions

A.E. designed and performed experiments and analysed data; A.L. and M.K. analysed data; Y.K, S.S., M.G, K.S-B and S.D. performed experiments; I.G.-T. designed and performed experiments, analysed data and wrote the manuscript; I.G.-T., E.B. and M.P. conceived ideas and oversaw the research. Inclusion and ethics statement: all authors in the locations where the research is conducted have been included as co-authors.

## Competing interests

The authors declare no competing interests.
