## [Transparent Peer Review file · Nature Communications]

Physiological and pathogenic autoreactivity converge in Type 1 Diabetes

Corresponding Author: Dr Iria Gomez-Tourino

Version 0:

Reviewer comments:

Reviewer #1

(Remarks to the Author)

The work as described provided a comprehensive examination of GAD65 specific CD4+ T cell responses in both healthy and T1D subjects. Approaches being used included ELISPOT, AIM assays, single cell gene expression of 48 selected genes and TCR study of GAD65 specific T cells. A study that provided a tremendous amount of data relating to autoreactive T cell responses to GAD65. Detail examinations of GAD65 TCR clonotypes in both HD and T1D subjects is the strength of this manuscript.

Major concerns

There are concerns related to the conclusions drawn from the gene expression study. The statistical test (Table 1) that used to show distributions of cells from HD and T1D in different clusters were difference is inappropriate. The Fisher's test as used assumes that the probability of each cell being from T1D or from HD is independent, and it clearly is not as many of those cells come from the same individuals. It's effectively treating the cell as the only level of measurement, and the subject from where the cell comes from is being ignored. It would be better to model the proportion of cells in each cluster in each individual and compare this between the two groups. The problem with the Fisher's test is that it hugely inflates the sample size, leading to false positives in the statistical tests. Results from the gene expression data also appeared to disagree with the AIM data, as clusters 1 and 2 are T1D enriched clusters and are enriched with naïve T cells (line 269), while the AIM assay show no difference in the naïve population between T1D and HD (Fig 1E)

Through TCR sequencing, they report the presence of GAD65 public TCR clonotype. The observation that the prevalences of public GAD65 specific TCR clonotypes in T1D and HD subjects in the naïve repertoires is interesting. Data show that GAD65 specific public TCR clonotypes from T1D were being detected in a higher percentage of naïve cell repertoires compared to those public TCR from HD. In addition, T1D GAD65 specific TCR clonotypes were frequent in naïve cells from HD, while HD GAD65 clonotypes were rare in naïve cells from T1D, implying HD GAD65 clonotypes could be protective. A critical piece of data that seems to be missing is to show that these public GAD65 specific TCR clonotypes are indeed GAD65 specific. Though these TCRs were derived from cells using the AIM assays, and these AIM+ cell lines were indeed GAD65 specific; there is no guarantee that every single AIM+ cells from the cell lines are indeed GAD65 specific. Re-expression of a few of these public clonotypes and demonstrate their specificity to GAD65 will add tremendous values to this manuscript.

Other minor points:

For Fig 3B, the data gave the impression that the lines are collection TCRs with identical aa. Is that the case? Or they simply belong to a convergence group with similar CDR3 amino acid residues.

For Fig. 4H, a total of 4 data points from the T1D group, too low a number to draw any conclusion.

Reviewer #2

(Remarks to the Author)

The manuscript by Gomez-Tourino and colleagues reports on studies using peripheral blood immune cells to evaluate physiologic versus pathogenic autoimmunity in type 1 diabetes. The studies are well done and provides a comprehensive

analysis of GAD reactive autoimmunity using several different assays including functional cytokine ELISPOT assays, activation-induced marker (AIM) assays for antigen specific CD4 T cells and paired single cell gene expression with T cell receptor sequencing for GAD-specific T cells. The research question has significant relevance to type 1 diabetes, many other autoimmune disorders, and the general field of immunology. The novelty of the work lies in the different assay approaches used to investigate the research question, along with the findings using tracked antigen specific T cell receptor sequences from T cell subsets back to bulk peripheral blood TCR sequencing. My major concerns are with the data analysis and interpretation of that data to support the overall conclusion of the work. My major critiques are listed below:

1. Introduction – please select terms such as physiologic and pathogenic autoimmunity and eliminate confusing and confounding terms such as “beneficial” autoimmunity.
2. The results section that analyzes GAD-specific CD4 T cells based upon gene expression profiles (Table 1 and Figure 2) does not show marked differences between individuals with and without T1D. Table 1 should report frequencies or proportions of cells as there are different inputs from T1D cells (n=912) compared to healthy donors (n=548). It is not apparent that there are T1D or HD enriched clusters in figure 2C – again the input numbers are quite different.
3. The text in the results section describing the 10 different clusters of GAD-specific cells based upon gene expression is dense and there are not notable differences between disease states.
4. Are there distinct differences in GAD-specific alpha/beta TCRs based upon gene expression profiles between T1D and HD cells?
5. Figure 5 is difficult to read and interpret. This may work better as a supplemental figure.
6. Can any insights be gained by looking at different GAD-specific cell nucleotide TCR sequences that converge to the same amino acid TCR sequence? This potentially indicates that there was in vivo expansion of these T cell clones, as opposed to multiple amplifications of the same DNA sequence during the PCR used for sequencing.
7. It is interesting and novel that specific TCR sequences are localized to T1D cells and separate sequences are in healthy donors (Figure 6 for both TN and CM subsets). What characteristics differentiate these T1D vs. HD TCR sequences in this dataset? Are different V genes used, CDR3 length, or hydrophobic amino acids within the TRBV CDR3?
8. The discussion and proposed model in Figure 7 need to acknowledge that there are other T cell targets outside of GAD (e.g., preproinsulin, IA-2, ZnT8, post-translationally modified islet antigens) that contribute to ‘diabetogenic’ autoreactivity.

Version 1:

Reviewer comments:

Reviewer #1

(Remarks to the Author)

The authors have addressed my previous concerns.

There are a two sentences in the results section in this revised version that are ambiguous or even contradictory. Please clarify those statements.

line 198-200:

We instead found T1D-specific differences in the expression levels of specific genes, suggesting that GAD-specific responses are of similar magnitudes as of HD ones, but of alter nature.

Reviewer #2

(Remarks to the Author)

My comments were adequately addressed, and I believe the overall manuscript has improved through the review process. Excellent manuscript!

RESPONSE TO REVIEWERS' COMMENTS

We thank the reviewers for their constructive comments on our manuscript “*Physiological and pathogenic autoreactivity converge in Type 1 Diabetes*”. We feel that we are able to respond to all of the Reviewers’ comments appropriately, including the revision of the statistical tests used, the development of transduction assays to confirm GAD specificity, and the introduction of new analyses, particularly on convergence and HD-only/T1D-only clonotypes.

A point-by-point response to both reviewers is shown below.

Reviewer #1 (Remarks to the Author):

The work as described provided a comprehensive examination of GAD65 specific CD4+ T cell responses in both healthy and T1D subjects. Approaches being used included ELISPOT, AIM assays, single cell gene expression of 48 selected genes and TCR study of GAD65 specific T cells. A study that provided a tremendous amount of data relating to autoreactive T cell responses to GAD65. Detail examinations of GAD65 TCR clonotypes in both HD and T1D subjects is the strength of this manuscript.

Thank you for your positive comments.

Major concerns:

R1_#01: *There are concerns related to the conclusions drawn from the gene expression study. The statistical test (Table 1) that used to show distributions of cells from HD and T1D in different clusters were difference is inappropriate. The Fisher’s test as used assumes that the probability of each cell being from T1D or from HD is independent, and it clearly is not as many of those cells come from the same individuals. It’s effectively treating the cell as the only level of measurement, and the subject from where the cell comes from is being ignored. It would be better to model the proportion of cells in each cluster in each individual and compare this between the two groups. The problem with the Fisher’s test is that it hugely inflates the sample size, leading to false positives in the statistical tests.*

Thank you for raising this issue, which has also been raised by Reviewer #2. We have repeated our statistical analysis following the approach suggested by both reviewers, and no statistical differences are found in the distribution of cells from HD and T1D patients among the clusters. Therefore, we have deleted Table 1, modified Figure 2C, described the statistical analysis in the Methods section, and modified the text in the Results and Discussion sections. Qualitative differences in gene expression between HD and T1D still remain significant, and we have kept those in the main text and relevant figures.

R1_#02: *Results from the gene expression data also appeared to disagree with the AIM data, as clusters 1 and 2 are T1D enriched clusters and are enriched with naïve T cells (line 269), while the AIM assay show no difference in the naïve population between T1D and HD (Fig 1E)*

After repeating the statistical analyses (former Table 1) with the appropriate statistical test, clusters #1 and #2 are not considered now as T1D-enriched clusters - we have modified the test accordingly, as explained in R1_#01. We still find, however, that the cell surface

phenotype of sorted cells analyzed for single-cell gene expression is different in T1D and HD cells, with T1D patients presenting higher proportions of TN cells, and lower proportions of EM cells (Figure 2J). This is partially in line with the results shown in Figure 1E, with T1D patients showing higher frequencies of EM cells. In this case, there are no differences in the naïve population, maybe due to the high scatter distribution of the individuals.

R1_#03: *Through TCR sequencing, they report the presence of GAD65 public TCR clonotype. The observation that the prevalences of public GAD65 specific TCR clonotypes in T1D and HD subjects in the naïve repertoires is interesting. Data show that GAD65 specific public TCR clonotypes from T1D were being detected in a higher percentage of naïve cell repertoires compared to those public TCR from HD. In addition, T1D GAD65 specific TCR clonotypes were frequent in naïve cells from HD, while HD GAD65 clonotypes were rare in naïve cells from T1D, implying HD GAD65 clonotypes could be protective. A critical piece of data that seems to be missing is to show that these public GAD65 specific TCR clonotypes are indeed GAD65 specific. Though these TCRs were derived from cells using the AIM assays, and these AIM+ cell lines were indeed GAD65 specific; there is no guarantee that every single AIM+ cells from the cell lines are indeed GAD65 specific. Re-expression of a few of these public clonotypes and demonstrate their specificity to GAD65 will add tremendous values to this manuscript.*

Thank you for the suggestion, as we agree that this type of validation would add value to our manuscript. We focused our efforts on TCRB clonotypes that were public at the nucleotide level in several T cell subsets (Figure 4A-D), as we think they represent extreme examples of publicity in our dataset. We selected two TCRB for which we had paired TCRA information (one from a healthy donor and one from a T1D patient), transduced them and confirmed their specificity against whole GAD, as indicated in the new Supplementary Figure 14. We feel that the combination of the specificity demonstrated by us and others of the AIM assay, plus the line generation results (Supplementary Fig. 4) and the new transduction experiments (Suppl. Fig 14) support the antigen specificity of these public clonotypes.

Other minor points:

R1_#04: *For Fig 3B, the data gave the impression that the lines are collection TCRs with identical aa. Is that the case? Or they simply belong to a convergence group with similar CDR3 amino acid residues.*

The lines in Figure 3B join clonotypes that belong to the same convergence group with similar CDR3 amino acid residues. We have made that point clearer in the Figure caption.

R1_#05: *For Fig. 4H, a total of 4 data points from the T1D group, too low a number to draw any conclusion.*

We have toned down the conclusions related to this Figure.

Reviewer #2 (Remarks to the Author):

The manuscript by Gomez-Trouirino and colleagues reports on studies using peripheral blood immune cells to evaluate physiologic versus pathogenic autoimmunity in type 1 diabetes. The studies are well done and provides a comprehensive analysis of GAD reactive autoimmunity using several different assays including functional cytokine ELISPOT assays, activation-induced marker (AIM) assays for antigen specific CD4 T cells and paired single cell gene expression with T cell receptor sequencing for GAD-specific T cells. The research question has significant relevance to type 1 diabetes, many other autoimmune disorders, and the general field of immunology. The novelty of the work lies in the different assay approaches used to investigate the research question, along with the findings using tracked antigen specific T cell receptor sequences from T cell subsets back to bulk peripheral blood TCR sequencing.

Thank you for your positive comments.

My major concerns are with the data analysis and interpretation of that data to support the overall conclusion of the work. My major critiques are listed below:

R2_#01: *1. Introduction – please select terms such as physiologic and pathogenic autoimmunity and eliminate confusing and confounding terms such as “beneficial” autoimmunity.*

We use the term “beneficial” once in the manuscript, between quotation marks in the Introduction section. This is the term employed in reference 6 (Hauben, E., Roncarolo, M. G., Nevo, U. & Schwartz, M. Beneficial autoimmunity in Type 1 diabetes mellitus. Trends Immunol 26, 248-253 (2005)), one of the first papers that suggested the existence of physiological autoimmunity in Type 1 Diabetes. Hence, we have kept that term between quotation marks to accurately reflect what the authors meant in their publication.

R2_#02: *2. The results section that analyzes GAD-specific CD4 T cells based upon gene expression profiles (Table 1 and Figure 2) does not show marked differences between individuals with and without T1D. Table 1 should report frequencies or proportions of cells as there are different inputs from T1D cells (n=912) compared to healthy donors (n=548). It is not apparent that there are T1D or HD enriched clusters in figure 2C – again the input numbers are quite different.*

Thank you for raising this issue, which has also been raised by Reviewer #1. We have repeated our statistical analysis following the approach suggested by both reviewers, and no statistical differences are found in the distribution of cells from HD and T1D patients among the clusters. Therefore, we have deleted Table 1, modified Figure 2C, described the statistical analysis in the Methods section, and modified the text in the Results and Discussion sections. Qualitative differences in gene expression between HD and T1D still remain significant, and we have kept those in the main text and relevant figures.

R2_#03: 3. *The text in the results section describing the 10 different clusters of GAD-specific cells based upon gene expression is dense and there are not notable differences between disease states.*

We have now significantly shortened this section, focusing the attention on T1D-specific features, namely gene expression differences. Accordingly, we have modified Supplementary Figures 9 and 10.

R2_#04: 4. *Are there distinct differences in GAD-specific alpha/beta TCRs based upon gene expression profiles between T1D and HD cells? Further explanation received by email to the editor: "I'm happy to clarify. I am interested in knowing if there are differences in the gene expression for the single cells in which they have paired alpha and beta TCRs (not those having a single TCR chain) between T1D and HD. Ideally, they could confirm that several of these CD4 T cells with paired alpha and beta chains are truly GAD specific."*

We prepared a new plot (Supplementary Figure 11B) to show this data and mention it in the Results section. No evident differences were found in the gene cluster distribution of HD and T1D cells having both TCRA and TCRB information.

Regarding the truly specificity of some cells, we selected two TCRB for which we had paired TCRA information (one from a healthy donor and one from a T1D patient), transduced them and confirmed their specificity against whole GAD, as indicated in the new Supplementary Figure 14. We feel that the combination of the specificity demonstrated by us and other of the AIM assay, plus the line generation results (Supplementary Fig. 4) and the new transduction experiments (Suppl. Fig 14) support the antigen specificity of these public clonotypes.

R2_#05: 5. *Figure 5 is difficult to read and interpret. This may work better as a supplemental figure.*

Apologies for the small type of font. We now display these plots in landscape, and split Figure 5 in two figures (now Figure 6 and Figure 7). We feel these should remain as main figures, as they depict the relationship between convergence and frequency.

R2_#06: 6. *Can any insights be gained by looking at different GAD-specific cell nucleotide TCR sequences that converge to the same amino acid TCR sequence? This potentially indicates that there was in vivo expansion of these T cell clones, as opposed to multiple amplifications of the same DNA sequence during the PCR used for sequencing.*

We would like to thank the reviewer for the suggestion, as we feel that these results strengthen the study. These new results are now detailed in the Results section, and in Figure 8 E-H. We found that T1D-derived GAD clonotypes that are expanded into CM subsets present a higher degree of convergence than HD-derived and CMV ones.

R2_#07: 7. *It is interesting and novel that specific TCR sequences are localized to T1D cells and separate sequences are in healthy donors (Figure 6 for both TN and CM subsets). What characteristics differentiate these T1D vs. HD TCR sequences in this dataset? Are different V genes used, CDR3 length, or hydrophobic amino acids within the TRBV CDR3?*

Thank you for suggesting this additional analysis, which completes the last section of the Results. We have now performed this study, as shown in the Results Section, Figure 9 E-G, Supplementary Figure 17 and Supplementary Tables 6 and 7. We observe that GAD clonotypes found exclusively in CM cells of HD donors have higher clonal sizes, and include regulatory/multiple TF phenotypes not present in T1D-only clonotypes. V gene and amino acid usages are also different between HD-only and T1D-only GAD clonotypes.

R2_#08: 8. *The discussion and proposed model in Figure 7 need to acknowledge that there are other T cell targets outside of GAD (e.g., preproinsulin, IA-2, ZnT8, post-translationally modified islet antigens) that contribute to 'diabetogenic' autoreactivity.*

We have modified Figure 7 (now Figure 10) to include the contribution of other T1D autoantigens and made reference to their role in the discussion of the model.

RESPONSE TO REVIEWERS' COMMENTS

We would like to thank both reviewers for their constructive comments, which have significantly improved the manuscript.

Reviewer #1 (Remarks to the Author):

The authors have addressed my previous concerns.

There are a two sentences in the results section in this revised version that are ambiguous or even contradictory. Please clarify those statements.

line 198-200:

We instead found T1D-specific differences in the expression levels of specific genes, suggesting that GAD-specific responses are of similar magnitudes as of HD ones, but of alter nature.

Thank you for pointing this out. We have rephrased this sentence as follows: “). However, we find T1D-specific differences in the expression levels of specific genes within specific clusters”.

Reviewer #2 (Remarks to the Author):

My comments were adequately addressed, and I believe the overall manuscript has improved through the review process. Excellent manuscript!

Thank you for your comments, and for the suggestions of the new analyses, which have improved the study.